# Pleomorphic effects of three small-molecule inhibitors on transcription elongation by *Mycobacterium tuberculosis* RNA polymerase

Omar Herrera-Asmat[1,2,3,4†], Alexander B Tong[1,4,5†], Wenxia Lin[4,6], Tiantian Kong[6], Juan R Del Valle[7], Daniel G Guerra[3], Yon W Ebright[8,9], Richard H Ebright[8,9], Carlos Bustamante[1,2,4,5,10,11,12,13]*

[1]Jason L. Choy Laboratory of Single-Molecule Biophysics, University of California, Berkeley, Berkeley, United States; [2]Department of Molecular and Cell Biology, University of California, Berkeley, Berkeley, United States; [3]Laboratorio de Moléculas Individuales, Laboratorios de Investigación y Desarrollo, Facultad de Ciencias e Ingeniería, Universidad Peruana Cayetano Heredia, Lima, Peru; [4]California Institute for Quantitative Biosciences, University of California, Berkeley, Berkeley, United States; [5]Department of Chemistry, University of California, Berkeley, Berkeley, United States; [6]Department of Biomedical Engineering, School of Medicine, Shenzhen University, Shenzhen, China; [7]Department of Chemistry & Biochemistry, University of Notre Dame, Notre Dame, United States; [8]Waksman Institute, Rutgers University, New Brunswick, United States; [9]Department of Chemistry and Chemical Biology, Rutgers University, New Brunswick, United States; [10]Department of Physics, University of California, Berkeley, Berkeley, United States; [11]Biophysics Graduate Group, University of California, Berkeley, Berkeley, United States; [12]Kavli Energy Nanoscience Institute, University of California, Berkeley, Berkeley, United States; [13]Howard Hughes Medical Institute, University of California, Berkeley, Berkeley, United States

*For correspondence:
carlosb@berkeley.edu

†These authors contributed equally to this work

**Competing interest:** The authors declare that no competing interests exist.

**Abstract** The *Mycobacterium tuberculosis* RNA polymerase (MtbRNAP) is the target of the first-line anti-tuberculosis inhibitor rifampin; however, the emergence of rifampin resistance necessitates the development of new antibiotics. Here, we communicate the first single-molecule characterization of MtbRNAP elongation and its inhibition by three diverse small-molecule inhibitors: N(α)-aroyl-N-aryl-phenylalaninamide (D-IX216), streptolydigin (Stl), and pseudouridimycin (PUM) using high-resolution optical tweezers. Compared to *Escherichia coli* RNA polymerase (EcoRNAP), MtbRNAP transcribes more slowly, has similar mechanical robustness, and only weakly recognizes *E. coli* pause sequences. The three small-molecule inhibitors of MtbRNAP exhibit strikingly different effects on transcription elongation. In the presence of D-IX216, which inhibits RNAP active-center bridge-helix motions required for nucleotide addition, the enzyme exhibits transitions between slowly and super-slowly elongating inhibited states. Stl, which inhibits the RNAP trigger-loop motions also required for nucleotide addition, inhibits RNAP primarily by inducing pausing and backtracking. PUM, a nucleoside analog of UTP, in addition to acting as a competitive inhibitor, induces the formation of slowly elongating RNAP-inhibited states. Our results indicate that the three classes of small-molecule inhibitors affect the enzyme in distinct ways and show that the combination of Stl and D-IX216, which both target the RNAP bridge helix, has a strong synergistic effect on the enzyme.

## Editor's evaluation

This important work used state-of-the-art optical tweezers to investigate the elongation of *Mycobacterium tuberculosis* RNAP (mtbRNAP) and its inhibition by three small-molecule inhibitors: N(α)-aroyl-N-aryl-phenylalaninamide (D-IX216), streptolydigin (Stl), and pseudouridimycin (PUM). The authors provide compelling evidence demonstrating a slower elongation mode of MtbRNAP compared to that of the *E. coli* RNAP, distinct inhibition modes of the three small molecule inhibitors, and a synergistic effect of Stl and D-IX216 on MtbRNAP. These findings clarified the differential mechanistic actions of RNA polymerase inhibitors, providing insights for the development of new, effective combination therapies against tuberculosis.

## Introduction

Tuberculosis (TB) is the second most prevalent infectious disease causing morbidity and mortality worldwide (after COVID-19). There are 10 million new cases and 1.5 million deaths yearly, which incur $12 billion in medical and economic costs (*World Health Organization, 2023*). Rifampin is a small-molecule inhibitor of MtbRNAP and is a first-line treatment for TB (*Severinov et al., 1994*; *Molodtsov et al., 2017*). The emergence and proliferation of rifampin resistance in *M. tuberculosis* have led to an effort to identify and develop new small-molecule inhibitors that effectively inhibit rifampin-resistant MtbRNAP (*Xu et al., 2018*; *Gill and Garcia, 2011*; *Stefan et al., 2018*; *Liu et al., 2018*). A thorough understanding of the activity of MtbRNAP and its interactions with these new inhibitors is critical for informing and facilitating inhibitor development against TB.

In the last three decades, single-molecule optical tweezers studies of *E. coli* RNAP (EcoRNAP) and eukaryotic RNAP II have provided key insights on transcription elongation (*Yin et al., 1995*; *Wang et al., 1998*), including the direct, real-time observation of single-base-pair stepping (*Chakraborty et al., 2017*; *Abbondanzieri et al., 2005*). In addition, these studies have revealed details about sequence-dependent pausing, factor-dependent pausing (*Neuman et al., 2003*; *Gabizon et al., 2018*), and backtracking (*Gabizon et al., 2018*; *Lisica et al., 2016*; *Shaevitz et al., 2003*; *Galburt et al., 2007*). However, single-molecule studies of MtbRNAP and its inhibition by small-molecule inhibitors have been scarce. Current studies have primarily been structural and bulk assays (*Chen et al., 2021*; *Boyaci et al., 2020*; *Boyaci et al., 2018*; *Chakraborty, 1979*; *Lin et al., 2018*; *Gulten and Sacchettini, 2013*; *Hubin et al., 2017*), with only one study by single-molecule methods addressing initiation (*Vishwakarma et al., 2018*).

Here, we present a study of transcription elongation by MtbRNAP and its inhibition by small-molecule inhibitors using high-resolution single-molecule optical tweezers. First, we characterize *in singulo* the transcription elongation of MtbRNAP alone, including its pause-free velocity and mechanical robustness. Next, we assess three small-molecule inhibitors that target the transcription elongation of MtbRNAP. These inhibitors bind to sites distinct from rifampin and remain effective despite polymerase mutations that confer resistance to rifampin. These inhibitors are D-IX216 (an N(α)-aroyl-N-aryl-phenylalaninamide) (*Ebright et al., 2015*; *Lin et al., 2017*), streptolydigin (Stl) (*Rinehart et al., 1963*; *Kyzer et al., 2005*; *Temiakov et al., 2005*; *Arseniev et al., 2023*; *Zorov et al., 2014*; *Tuske et al., 2005*), and pseudouridimycin (PUM) (*Maffioli et al., 2019*; *Cain et al., 2022*; *Maffioli et al., 2017*).

N(α)-aroyl-N-aryl-phenylalaninamides selectively inhibit Mycobacterial RNAPs (*Lin et al., 2017*) by binding to a site at the N-terminal end of the bridge helix. This site is conserved in Mycobacterial polymerases but not in other bacterial polymerases. The bound inhibitor interferes with the conformational dynamics of the bridge helix that are necessary for nucleotide addition (*Ebright et al., 2015*; *Lin et al., 2017*; *Feng et al., 2015*; *Bae et al., 2015*). In this study, we used D-IX216, we found that addition of this inhibitor results in the enzyme transitioning to different inhibition states that add nucleotides much more slowly. Stl acts as a broad-spectrum inhibitor of bacterial RNAPs (*Rinehart et al., 1963*; *Kyzer et al., 2005*; *Temiakov et al., 2005*; *Zorov et al., 2014*; *Tuske et al., 2005*). Stl binds to the region between the bridge helix and trigger loop, interfering with the closing of the trigger loop, which is required for nucleotide addition (*Temiakov et al., 2005*; *Tuske et al., 2005*). We find that Stl stabilizes pausing by the enzyme. PUM is another broad-spectrum inhibitor of bacterial RNAPs (*Maffioli et al., 2019*) and is a nucleoside analog of UTP (*Maffioli et al., 2019*; *Cain et al., 2022*; *Maffioli et al., 2017*; *Böhringer et al., 2021*; *Sosio et al., 2018*), competitively inhibiting the

binding of UTP. Additionally, PUM also appears to have an additional inhibitory effect on the enzyme (*Maffioli et al., 2017*). Finally, when combined, D-IX216 and Stl exhibit synergistic inhibition of the enzyme.

## Results

### Biophysical and mechanochemical characterization of MtbRNAP

We studied the elongation of MtbRNAP using a high-resolution optical tweezers assay, where the RNAP and one end of the DNA template were tethered to polystyrene microspheres held in optical traps (*Figure 1a*, left panel). We collected single-molecule trajectories of MtbRNAP transcription across a ~3 kb template with a sequence derived from *M. tuberculosis*.

Molecular trajectories were first obtained in *assisting constant force* mode (*Figure 1b*), where the upstream DNA is tethered, causing the force applied by the traps to aid the movement of the RNAP. We also obtained trajectories in *opposing constant force* mode (*Figure 1c*), where the downstream DNA is tethered, and the force hinders the enzyme's movement. In both cases, as the RNAP moves, the length of the tether changes, and the position of the movable trap is adjusted via force feedback to maintain a constant tension on the tether.

Transcription by MtbRNAP under an assisting constant force of 18 pN comprises periods of continuous activity at a consistent velocity interspersed with long pauses, a behavior also described for EcoRNAP (*Adelman et al., 2002*; *Mejia et al., 2008*; *Figure 1a*, right panel). Plots of the velocity distribution of all molecular trajectories for both enzymes show a population at zero velocity corresponding to transcriptional pauses, and another corresponding to the pause-free velocity of the enzyme (*Figure 1d*, right panel). The pause-free velocity of MtbRNAP is measured by this method is 23.1±5.1 nt/s, approximately half that of EcoRNAP, which is 42.7±8.6 nt/s (*Figure 1d*, right panel).

To get a more precise characterization of the dynamics of RNAP elongation, we obtained the dwell time at each nucleotide by fitting the molecular traces to a monotonic one base-pair (bp) staircase function. We then modeled the resulting dwell time distribution (DTD) as a sum of exponentials ($\sum_{i=1}^{n} a_i e^{-k_i t}$) *Janissen et al., 2022*. We interpret these different exponentials as distinct states of the enzyme: $a_i$ represents the state probability ($\sum_{i=1}^{n} a_i = 1$), while $k_i$ indicates the kinetic rate of the *ith* state, such that $k_1 > k_2 > \ldots k_n$. The fastest state typically corresponds to the pause-free state, whereas the slower states reflect various dwell times associated with the paused states of the enzyme, including the relatively short elemental pauses and longer backtracking pauses.

For EcoRNAP, three exponential components effectively describe the DTD. The majority of the steps (92.4±1.3%, 95% CI), belong to the fastest state (state 1), with a rate of 38.3±1.0 nt/s ($k_{pause-free}$ [$k_{pf}$], 95% CI), which corresponds to the average rate of completing a pause-free nucleotide addition cycle (*Janissen et al., 2022*). State 2, comprising 7.2±1.3% (95% CI) of steps, corresponded to a slower rate of 6.2±1.0 nt/s ($k_{elemental-pause}$ [$k_{ep}$], 95% CI), which we associate with the rate of escape by RNAP from elemental pauses (*Keller and Bustamante, 2000*). Finally, the remaining 0.31±0.16% (95% CI) of steps are state 3, which corresponds to an even slower rate of 0.34±0.15 nt/s (95% CI), reflecting long pauses ($k_{lp}$).

For MtbRNAP, we also identified three populations with a $k_{pf}$ of 20.7±0.4 nt/s, a $k_{ep}$ 5.2±0.6 s$^{-1}$, and a $k_{lp}$ of 0.52 s$^{-1}$ with similar population percentages to EcoRNAP (*Figure 1e*). The pause-free velocity values obtained from the velocity distribution (*Figure 1d*, right panel) and DTD (*Figure 1e*) coincide well. We repeated this characterization for MtbRNAP under various magnitudes of assisting constant forces, ranging from 8 pN to 18 pN, and observed that $k_{pf}$ slightly increases with greater force, while $k_{el}$ was mostly unchanged. Conversely, when conducting the experiment under opposing constant forces from 4 to 18 pN, there are no dramatic effects on $k_{pf}$ or $k_{el}$ (*Figure 2a*).

Next, we performed experiments in *opposing passive mode*, where the downstream DNA is tethered, as in opposing constant force mode, but the traps are not moved, leading to an increase in force as transcription progresses (*Figure 2b*). Opposing passive mode is used to determine the stalling behavior and stall force of the polymerase. At low forces, MtbRNAP transcription velocity exhibits minimal response to the opposing force until a maximum force ($F_{stall}$) is reached, beyond which the enzyme drastically slows, stalls, and backtracks (*Figure 2b*; *Keller and Bustamante, 2000*). On average, MtbRNAP stalled at 19.3±1.2 pN (s.e.m.) (*Bustamante et al., 2004*). This force-induced

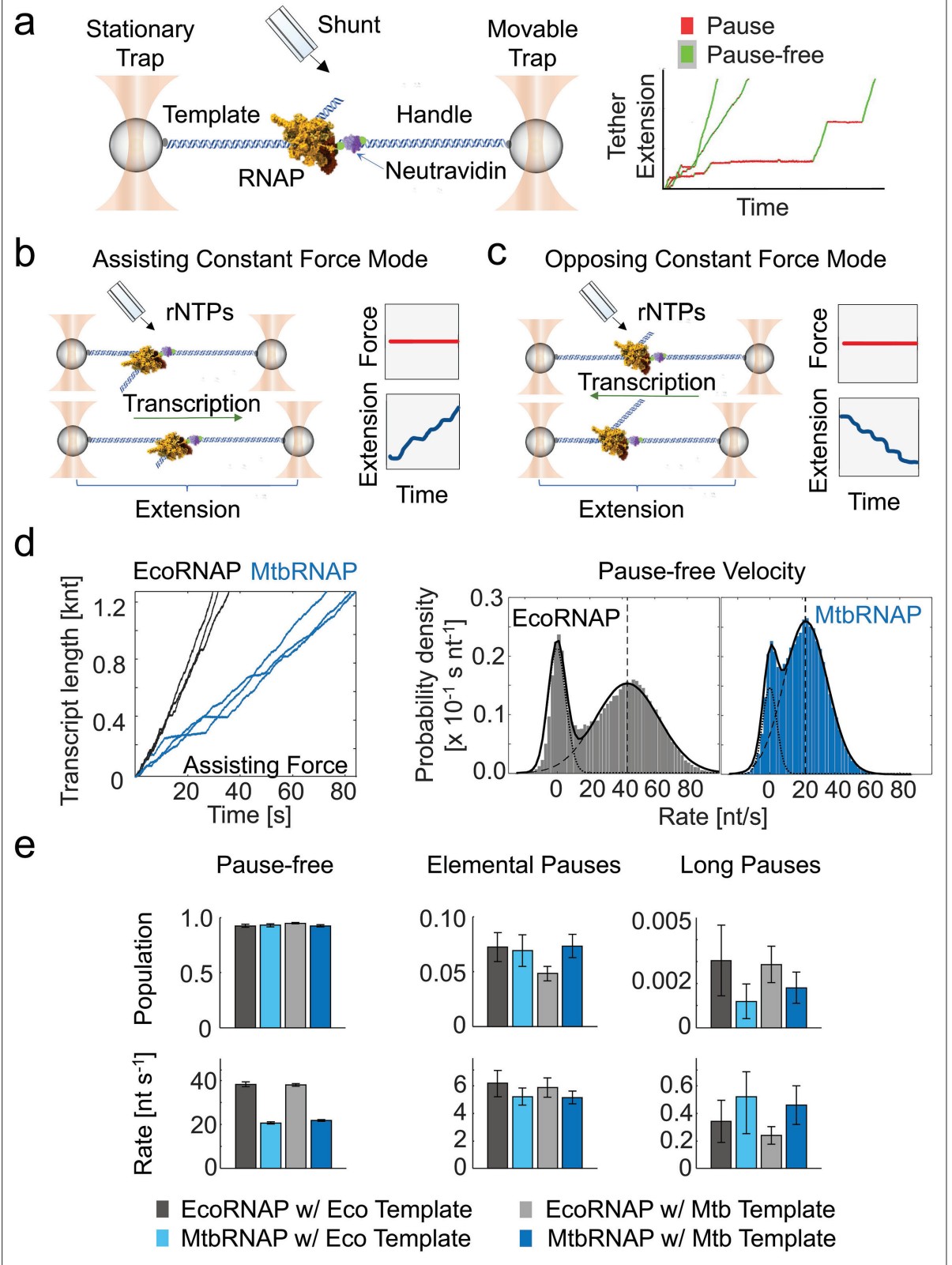

**Figure 1.** Single-molecule transcription elongation by *M. tuberculosis* RNA polymerase under high-resolution optical tweezers. (**a**) (left) The experimental optical tweezer setup is shown, which tethers and isolates the RNA polymerase (RNAP) from the laser beams through a DNA handle and template. (right) A few single-molecule transcription traces are shown, with the pauses (red) and the pause-free periods (green) highlighted. (**b**) A diagram of the assisting constant force mode is shown. The traps are moved apart as the RNAP transcribes to keep the force constant. (**c**) A diagram of

*Figure 1 continued*

the opposing constant force mode is shown. The traps are moved closer as the RNAP transcribes to keep the force constant. (**d**) (left) Representative trajectories of transcription by *Escherichia coli* RNA polymerase (EcoRNAP) (black) and *Mycobacterium tuberculosis* RNA polymerase (MtbRNAP) (blue) under ~18 pN assisting constant force and saturating concentration of rNTPs (~1 mM) are shown. The DNA template for MtbRNAP derives from *M. tuberculosis* genes *rpoB* and *rpoC*. (right) Velocity distributions of these trajectories are shown, with a fit to a sum of two Gaussians, one with mean zero (pauses) and one with positive mean (pause-free velocity). (**e**) The populations and rates obtained via dwell time distribution (DTD) analysis are shown for EcoRNAP and MtbRNAP on Eco and Mtb templates. Error bars are 95% CIs of fits of steps obtained from (left to right) 16, 19, 24, and 16 molecules. The traces were collected under saturating concentrations of nucleotides and 15–20 pN of assisting constant force.

stalling is often accompanied by backtracking (*Figure 2c*). By comparison, EcoRNAP stalls at 17.1±1.1 pN (s.e.m.) and is also accompanied by backtracking, consistent with previous studies (*Wang et al., 1998*; *Galburt et al., 2007*; *Kang et al., 2018*). The force-velocity curves can be fit to an unre-strained Boltzmann relation $v(f)=(1+A)/(1+Aexp(f\delta/kT))$, where the process is split into biochemical (force-independent) and mechanical (force-dependent) terms. The ratio of these rates at zero force is the parameter A, and the strength of force dependence is $\delta$ (*Figure 2d*, see Methods) (*Abbondanzieri et al., 2005*). For both Eco and Mtb RNAPs, we find that A<<1, indicating that biochemical transitions are rate-limiting at zero force in these instances, in line with prior findings for EcoRNAP (*Wang et al., 1998*). The value of δ is the distance to the transition state of polymerase stalling via backtracking, so we interpret this as the backtracking distance required by the RNAP to stall (*Bustamante et al., 2004*). For MtbRNAP, we obtain a value of 14±1 nt (95% CI) (*Figure 2d*), comparable to the 13±1 nt observed for EcoRNAP and findings from earlier studies on EcoRNAP (*Wang et al., 1998*).

## MtbRNAP weakly recognizes pause sequences of EcoRNAP

We have observed that MtbRNAP tends to pause less frequently than EcoRNAP under an assisting constant force on both the Eco and Mtb templates (*Figure 1e*). To gain a clearer understanding of the pausing dynamics, we utilized a *molecular ruler* template to compare the pausing behavior in a sequence-dependent manner. A molecular ruler consists of a tandem repeat of a sequence harboring one or more strong pauses (*Figure 3a*; *Herbert et al., 2006*). Because the trajectories of the different traces obtained on our high-resolution optical tweezers instrument all display the repeating pattern of pauses, it is possible to align them and determine the precise position of the RNAP on the template with nearly single-base pair resolution (*Gabizon et al., 2018*; *Chen et al., 2019*). Without a ruler, uncertainties in position caused by bead size variations or calibration errors cause the location of RNAP on the template to be not precisely known. We previously developed a ruler for EcoRNAP featuring five pauses designated as *a*, *b*, *c*, *d*, and *his*. However, we discovered that this ruler was inef-fective for MtbRNAP, which does not pause strongly enough at those locations, as seen in the heights of the peaks of the residence time histogram (RTH) of the repeat (*Figure 3b and c*). We confirmed this behavior using a gel assay, which shows different and often weaker pausing by MtbRNAP at these locations (*Figure 3d*). Notably, MtbRNAP demonstrates reduced recognition of the elemental pauses *a*, *b*, and *d* and surprisingly did not strongly recognize the strong hairpin pause sequence *his* from *E. coli*.

Since MtbRNAP recognized pause *c* the strongest (*Figure 3d*), we designed a new molecular ruler based on this sequence. The *Mtb* molecular ruler consists of eight repeats of the elemental pause *c*, inserted within a fragment of the *M. tuberculosis rpoB* gene. MtbRNAP pauses strongly at the expected location of 50 nt into the 68 nt repeat that makes up the ruler, with minor pauses at positions 5 and 30 nt in the repeat region (*Figure 3—figure supplement 1*). Using this tool, we proceeded to study the effects of the inhibitors in a sequence-dependent manner.

## D-IX216 inhibits elongation by MtbRNAP by switching it into interchangeable slow and super-slow inhibited states of transcription elongation

We investigated the effects of D-IX216 on the elongation activity of MtbRNAP. D-IX216 is an analog of D-AAP1 (*Lin et al., 2017*), having a 2-thiophenyl group in place of a phenyl group as ring A and a 5-fl uoro-2-piperazin-1-yl-phenyl instead of 2-methyl-phenyl as ring C (*Figure 4—figure supplement 1a*). D-IX216 is ten times more potent (IC$_{50}$ of 100 nM determined via bulk transcription methods) against

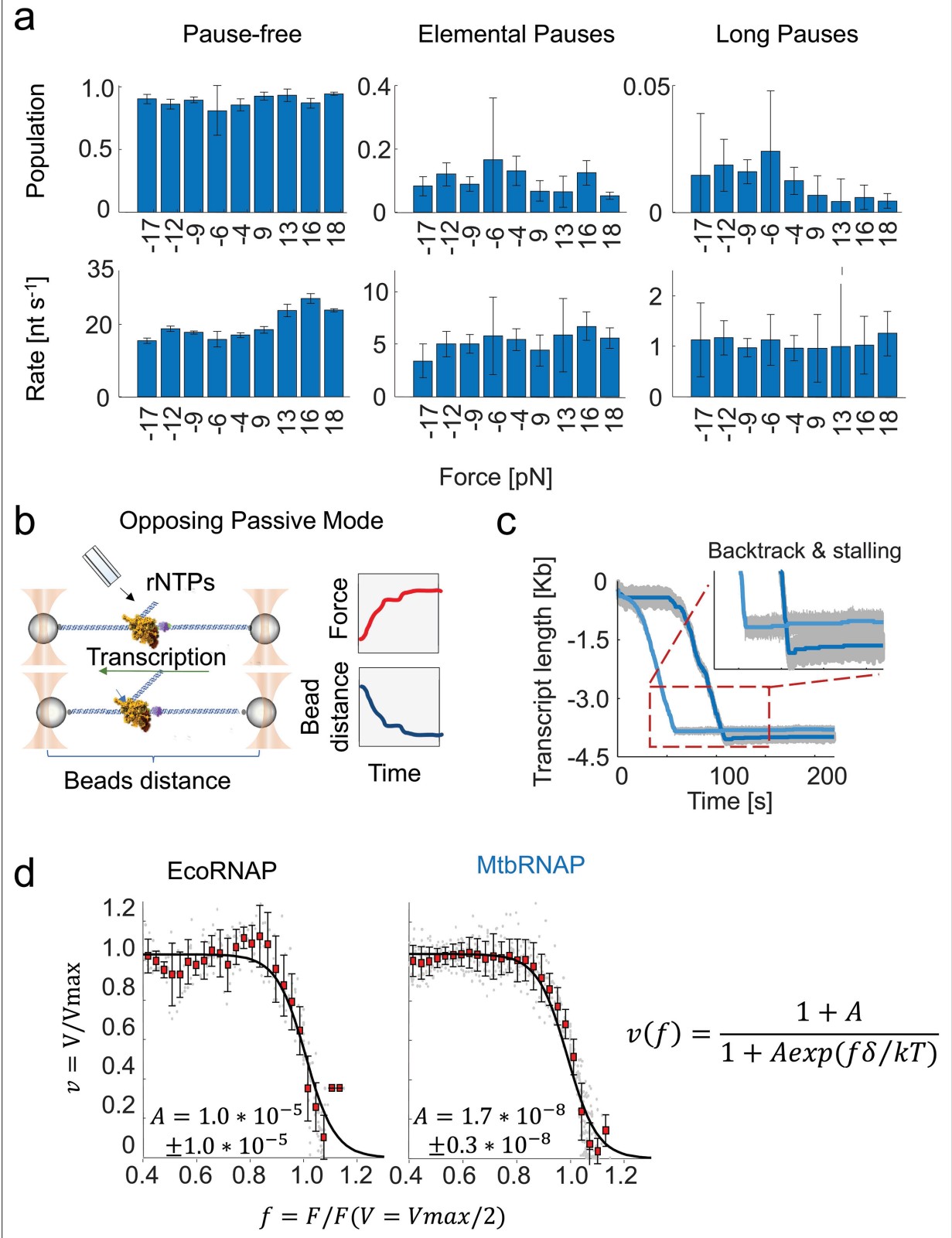

**Figure 2.** Mechanochemical characterization of the elongation by *M. tuberculosis* RNA polymerase under high-resolution optical tweezers. (**a**) The effect of magnitude and direction of applied force on the kinetic parameters obtained from dwell time distribution (DTD) analysis is shown. Positive forces are assisting constant forces, negative are opposing constant forces. Error bars are 95% CIs of fits of steps from (left to right) 4, 6, 10, 17, 14, 9, 3, 4, and 5 molecules. (**b**) A diagram of opposing passive mode is shown. The traps are held stationary as the RNA polymerase (RNAP) transcribes, leading

*Figure 2 continued*

to an increase in force. (**c**) Example opposing passive mode traces are shown. The *Mycobacterium tuberculosis* RNA polymerase (MtbRNAP) transcribes at near constant velocity before suddenly stalling and backtracking. (**d**) Force-velocity relationships for MtbRNAP and *Escherichia coli* RNA polymerase (EcoRNAP) obtained from passive mode experiments are shown and fit to a generalized Boltzmann relation (right). Both polymerases' velocities remain insensitive to force until they reach similar stall forces, on average 19.3 pN ± 1.2 pN for MtbRNAP, and 17.1 ± 1.1 pN for EcoRNAP. Error bars are standard deviations and come from 14 (Eco) and 22 (Mtb) molecules.

MtbRNAP compared to D-AAP1 (IC$_{50}$ of 1 µM) (*Ebright et al., 2015*; *Lin et al., 2017*), and is selective for MtbRNAP, not affecting EcoRNAP at these concentrations (*Figure 4—figure supplement 1b-d*).

Upon adding 70–560 nM D-IX216 to a transcribing MtbRNAP (*Figure 4a*) under 18 pN of assisting constant force, the elongation trajectories exhibited periods of transcription with three distinct global (i.e. including pauses) transcription velocities: one with *fast*, one with *slow*, and one with *super-slow* activity (*Figure 4b*), each lasting for at least tens of seconds (*Figure 4—figure supplement 1e–f*, and *Figure 4—figure supplement 2a*). The slow and super-slow were about 5 and 50 times slower than the fast, respectively. Additionally, the super-slow exhibited less than half the processivity (both in terms of the number of base pairs transcribed before tether break/detachment and in terms of the number of base pairs transcribed before recovery from inhibition) compared to the slow activity (*Figure 4—figure supplement 2b*). Interestingly, changing the direction of the applied force did not greatly alter the interconversion rate between these activities (*Figure 4—figure supplement 2b, c*).

The fast activity resembled the enzyme's behavior in the absence of D-IX216. DTD analysis of this activity yields values similar to D-IX216-free MtbRNAP, further suggesting that the fast activity corresponded to a D-IX216-free enzyme (*Figure 4—figure supplement 3*). Hence, we identify the fast DTD states as the pause-free, elemental pause, and the long pause states. In contrast, the slow and super-slow activities were exclusively observed with D-IX216 present, indicating that they arise from its binding to the RNAP. We shall refer to the slow and super-slow as *inhibited states* of the enzyme, to differentiate them from the nucleotide addition states obtained from DTD analysis mentioned earlier. When bound to D-IX216, MtbRNAP displayed a consistent slow nucleotide addition rate lasting tens of seconds (*Figure 4a and b*). The inhibited states persist for several steps and can interconvert with each other (*Figure 4—figure supplement 2b, c*). DTD analysis of these inhibited states revealed that the slow and super-slow inhibited states consist of multiple (nucleotide addition) states, each exhibiting distinct nucleotide addition rates (*Figure 4c and d*). We emphasize here that inhibited states refer to a persistent activity of the enzyme lasting for multiple incorporation cycles, while nucleotide addition states may change between every step. The rates of the various nucleotide addition states found for the slow and super-slow inhibited states do not match those of the pause-free, elemental pause, or long pause states (*Figure 4d*), so we conclude that these are states distinct from the normal ones visited by the polymerase.

When the concentration of D-IX216 was increased from 70 to 560 nM, the kinetic parameters extracted from the DTD analysis did not greatly change (*Figure 4—figure supplement 4a*), suggesting that the slow and super-slow inhibited states occur during the prolonged binding of a single D-IX216 molecule. This inference was supported by our washing experiments, which revealed that removing D-IX216 from the transcription medium of the affected polymerase did not immediately restore normal enzyme activity (*Figure 4—figure supplement 1f*). If this interpretation is correct, the conversion to the slow activity results from a binding event, while reversion to the fast activity is due to unbinding. Using the lifetimes of the fast and slow activities in the 280 nM case, we calculated a $k_{on}$, $k_{off}$, and $K_D$ for D-XI216, obtaining 0.133±0.006 µM$^{-1}$s$^{-1}$, 0.0032±0.0014 s$^{-1}$, and 24.1±10.6 nM, respectively (*Figure 4—figure supplement 4b*). The $K_D$ so attained is consistent with the IC$_{50}$ of ~100 nM. We have no explanation for what process is involved in the interconversion between the slow and the super-slow inhibited states, but we speculate that it may correspond to two distinct binding modes of the inhibitor to the enzyme, affecting its activity differently.

We also examined whether the induced slowing by D-IX216 could be due to reduced rNTP affinity by decreasing the rNTP concentration from 250 µM to 25 µM (~5 to ~0.5 x K$_M$ for the uninhibited polymerase) in the presence of D-IX216. However, the DTD behavior of the slow and super-slow inhibited states remained unchanged (*Figure 4—figure supplement 4c*), suggesting that rNTP binding is not rate-limiting in these inhibited states, unlike what is expected at these concentrations in the absence of the inhibitor.

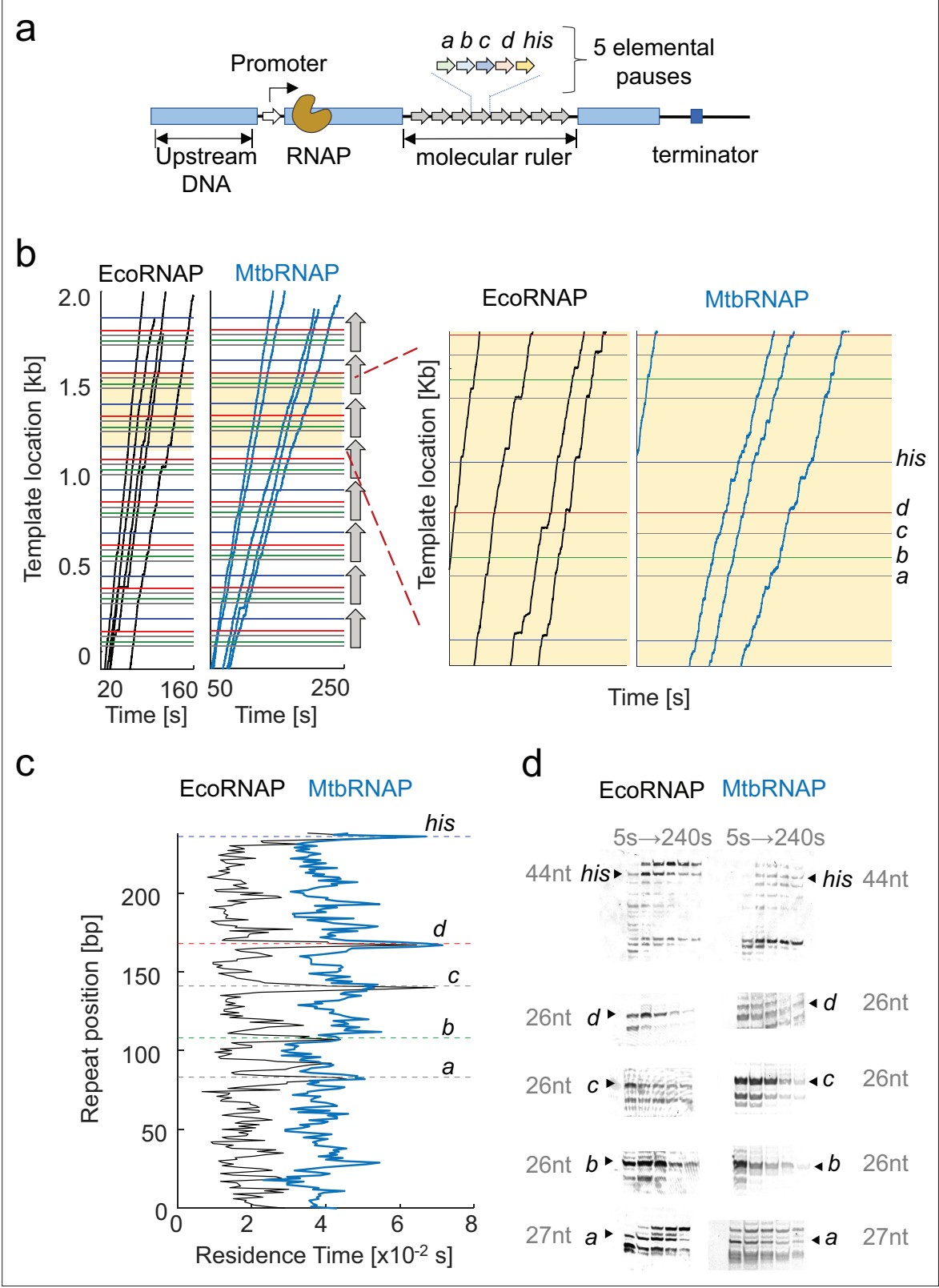

**Figure 3.** *Mycobacterium tuberculosis* RNA polymerase (MtbRNAP) pauses less efficiently than *Escherichia coli* RNA polymerase (EcoRNAP) on Eco pauses. (**a**) Representation of the template containing a molecular ruler to study EcoRNAP pauses. The DNA template consists of eight repeats of five Eco elementary pauses, *a, b, c, d*, and *his*. (**b**) Example traces of single Eco and Mtb RNA polymerases transcribing the *E. coli* molecular ruler are shown. The eight repeats are denoted as vertical arrows, and the location of the pauses within each repeat are shown as horizontal colored lines. We collected

*Figure 3 continued on next page*

*Figure 3 continued*

the transcription activities using a cocktail of rNTPs (1 mM rUTP, 1 mM rGTP, 0.5 mM rATP, and 0.25 mM rCTP) at 15–20 pN for EcoRNAP and 18 pN for MtbRNAP. The zoom on the right shows the alignment of the trace pauses with the expected pause locations. (**c**) Average residence time histograms of EcoRNAP and MtbRNAP display distinctive patterns of pause strength (peak height) and location. MtbRNAP more weakly recognizes the Eco pauses. (**d**) Comparison of the transcription gel bands at the elemental pause sites between MtbRNAP and EcoRNAP.

The online version of this article includes the following source data and figure supplement(s) for figure 3:

**Source data 1.** Original raw gels for *Figure 3d*.

**Source data 2.** Full gel images for *Figure 3d*, with labels.

**Figure supplement 1.** The design of a molecular ruler to study pausing in *Mycobacterium tuberculosis* RNA polymerase (MtbRNAP) with high-resolution optical tweezers.

**Figure supplement 1—source data 1.** Original raw gels for *Figure 3—figure supplement 1b*.

**Figure supplement 1—source data 2.** Full gel images for *Figure 3—figure supplement 1b*, with labels.

Lastly, we investigated the effects of D-IX216 on transcription across the Mtb molecular ruler to determine if these effects were sequence-dependent. Our findings revealed that slow or super-slow events occurred randomly along the DNA template, consistent with our observations using the original sequence of Mtb DNA (*Figure 4a*). When MtbRNAP is in the slow inhibited state while crossing the repeats region, we observed that D-IX216 increases the residence time at every position (*Figure 4e*). We normalized the RTH values to the median residence time of each inhibited state to account for the different incorporation times associated with the fast, slow, and super-slow inhibited states (*Figure 4e*, middle). This normalization demonstrated that the relative strength of the designated pause increased, as indicated by the heightened RTH peak (*Figure 4e*, right). If pause *c* is an off-pathway pause, a kinetic competition should exist between forward translocation and pausing. Therefore, we attribute the increased strength of this pause to a decrease in the transcription rate caused by the inhibitor. In conclusion, we found that D-IX216 globally slows MtbRNAP translocation in a sequence-independent manner.

## Streptolydigin induces backtracking in MtbRNAP

Using the Mtb molecular ruler, we added 4.5–45 µM Stl to a transcribing MtbRNAP under 18 pN of assisting constant force. We observed that Stl induced long pauses both inside and outside the repeat region (*Figure 5a*). The probability of entering these long pauses (lasting over 1 s) increased from 0.42% without the inhibitor to 0.76% with Stl. Of the enzymes that recovered, the long pause consists of two components; first, the RNAP pauses with an average duration of 10±1.3 s (95% CI); then, it backtracks by 7±1 nt (s.e.m.) and spends an additional 32±4 s (95% CI) paused before transcription resumes (*Figure 5b*). In contrast, previous studies have shown that Stl does not affect the backtracking dynamics of EcoRNAP (*Arseniev et al., 2023*; *Zorov et al., 2014*; *Tuske et al., 2005*).

We also conducted a real-time bulk fluorescence assay where a fluorescent adenosine analog 2-aminopurine (2AP) in the template strand changes its brightness depending on whether it is base-paired or not. In this assay, we assemble the RNAP bubble to be just in front of the 2AP, whereby translocation by the RNAP causes the 2AP to un-basepair and fluoresce more strongly. The cognate rNTP is not present in solution, so the fluorescence persists after translocation. This allows us to distinguish between the post-translocated and the other states of the enzyme (pre-translocated or otherwise, see *Figure 5—figure supplement 1a*). We used this assay to monitor the translocation state of RNAPs without cognate nucleotide and in the presence of 7.5 µM Stl (*Malinen et al., 2014*; *World Health Organization & World Health Organization, 2010*). These experiments were conducted without pyrophosphate to prevent the inherent editing of the enzyme in the backtracked state. In the absence of Stl, the MtbRNAP is predominantly found in the post-translocated state, while in the presence of Stl, only ~20% of RNAPs remained in the post-translocated state (*Figure 5—figure supplement 1b*). This reduction in the amount of post-translocated MtbRNAPs with Stl is consistent with the increase in backtracking observed in the single molecule traces (*Figure 5b*).

The recovery from the Stl-induced pauses before and after backtracking both exhibited single-exponential distributions (*Figure 5c*). A single-exponential distribution of backtrack durations is associated with *deep backtracking* and recovery via endonucleolysis, whereas a power law distribution ($t^{-3/2}$) corresponds to diffusive recovery from more shallow backtracking (*Figure 5d*).

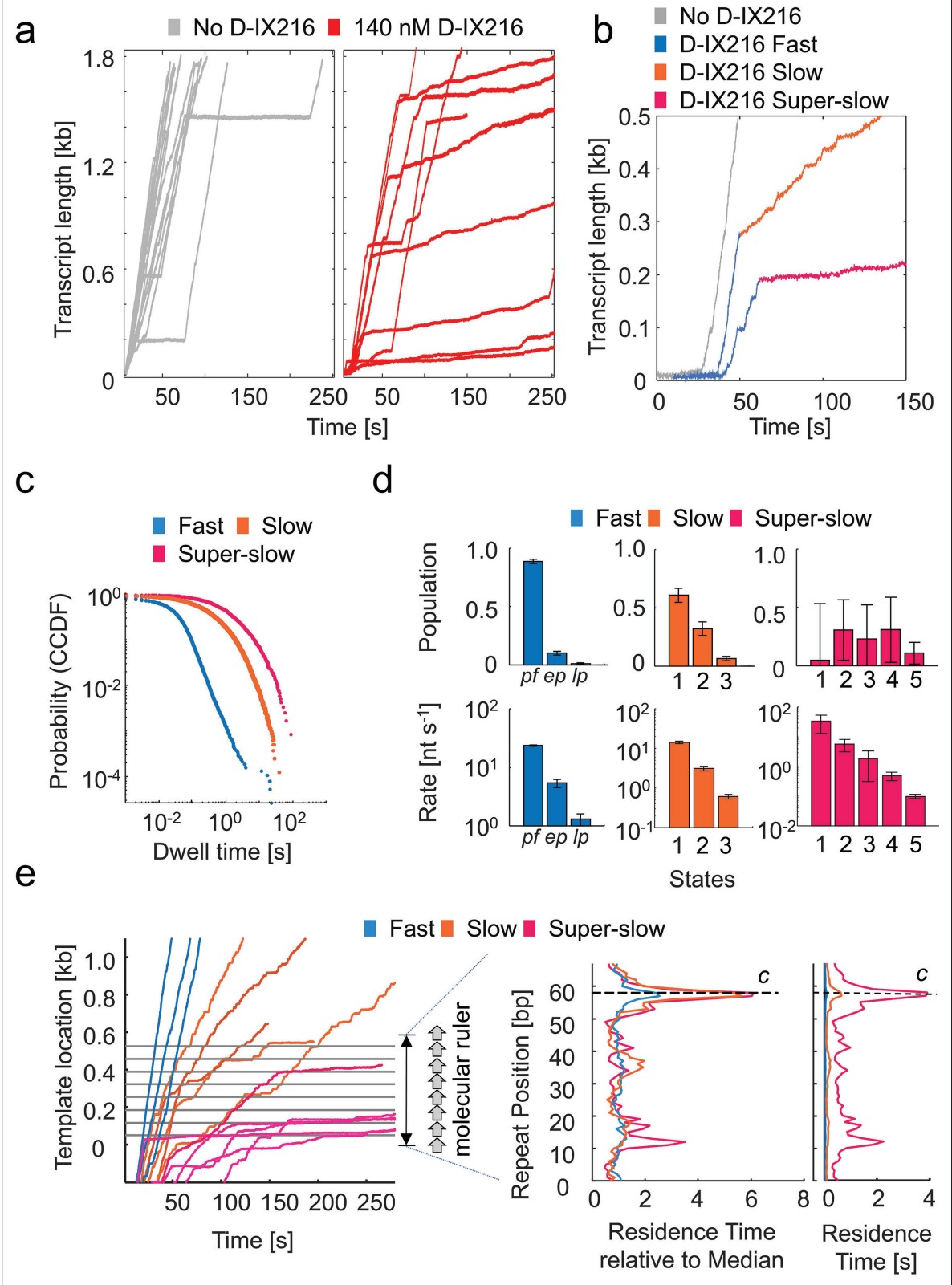

**Figure 4.** The inhibition of *Mycobacterium tuberculosis* RNA polymerase (MtbRNAP) by D-IX216 involves the conversion to one of two slowly elongating inhibited states. (**a**) Example traces of MtbRNAP transcribing in the absence (gray) and presence (red) of 140 nM D-IX216 are shown. (**b**) Traces exhibiting the three states, fast, slow, and super-slow, are shown and the regions are colored blue, orange, and red, respectively. Transcription in the absence of D-IX216 is shown in gray. The traces plotted in this figure were collected at 18 pN of assisting constant force, 1 mM rNTPs, and 140 nM D-IX216.

*Figure 4 continued on next page*

*Figure 4 continued*

(**c**) Comparison of the complementary cumulative distribution function (CCDF) of the dwell time distributions (DTD) of the fast, slow, and super-slow states. (**d**) DTD analysis of the fast, slow, and super-slow inhibited states are shown. Pf: pause-free, ep: elemental pause, lp: long pause. Error bars are 95% CIs of fits of steps obtained from 60 molecules. (**e**) (left) MtbRNAP traces crossing the Mtb molecular ruler are shown in varying states of inhibition. (right) The residence time histograms of these inhibited states are shown both in seconds or normalized to the median residence time.

The online version of this article includes the following figure supplement(s) for figure 4:

**Figure supplement 1.** D-IX216 specifically slows down the *Mycobacterium tuberculosis* RNA polymerase (MtbRNAP) transcription.

**Figure supplement 2.** Interconversion between the slow and super-slow inhibited states.

**Figure supplement 3.** The fast elongating state of *Mycobacterium tuberculosis* RNA polymerase (MtbRNAP) observed in the presence of D-IX216 corresponds to the inhibitor-free state.

**Figure supplement 4.** Effect of different D-IX216 and nucleotide concentrations on the slow and super-slow inhibited states of *Mycobacterium tuberculosis* RNA polymerase (MtbRNAP).

Next, we compared the effect of Stl on the DTD under assisting and opposing constant forces. We found that Stl did not affect the MtbRNAP pause-free velocity under assisting constant force (*Figure 5e*). However, under opposing constant force, which favors the pre-translocated state of the polymerase, Stl slightly reduced the polymerase velocity and greatly increased the occurrence and duration of long pauses compared to the enzyme without the inhibitor (*Figure 5e*), consistent with reports regarding Stl's effect on EcoRNAP (*Arseniev et al., 2023*).

Our examination of the RTHs across the molecular ruler by MtbRNAP under assisting constant force with and without Stl revealed that this inhibitor increased the duration of the designed pause in the tandem repeat region by 30%, but did not introduce additional sequence-dependent pause sites (*Figure 5f*). This observation suggests that Stl binding is more favorable when the enzyme is paused (*Temiakov et al., 2005*). This inference is supported by the fact that we did not observe an increased residence time background in the presence of Stl (outside of repeat pauses).

A recent article noted that factors GreA and GreB enhanced Stl's pausing effects on EcoRNAP under opposing force (*Arseniev et al., 2023*). However, when we assayed MtbRNAP elongation under comparable conditions using GreA (as GreB has not been found in Mtb), we observed no obvious change in long pause duration (*Figure 5g*). These differing results may reflect differences in the mechanism of Stl inhibition between EcoRNAP and MtbRNAP.

## Pseudouridimycin converts MtbRNAP into two slowly elongating inhibited states and increases the density of pauses in EcoRNAP

When 1 µM PUM was added to a transcribing EcoRNAP, pauses lasting 5.9±1.8 s on average were observed (pause escape rate of 0.17±0.04 s$^{-1}$), representing 2% of the enzyme's steps (*Figure 6a*). Occasionally, we noted longer pauses lasting several hundreds of seconds. Additionally, we rarely (in less than 6% of traces) observed sections where EcoRNAPs converted to one of two states of slower nucleotide addition (slow and super-slow). These states resemble those caused by D-IX216 on MtbRNAP, so we shall also refer to them as inhibited states; however, the enzyme was not observed to recover (*Figure 6b and c*).

Using the molecular ruler method with EcoRNAP, our experiments showed that PUM increased the RTH peaks of the *E. coli* native pauses *his* and *d*, reduced the RTH peak of native pause *c*, and sporadically introduced more sequence-dependent pauses along the template, and in a concentration-dependent manner (*Figure 6d*). As PUM is an rUTP analog, it is reasonable to conclude that these additional sequence-dependent pauses occur at U incorporation sites, as it seems to be the case for the additional pausing around the *his* and *d* sites (*Figure 6d*). However, due to the ±3 bp resolution of our technique, we cannot confidently assert that all of these new peaks correspond to U incorporation sites (*Shaevitz et al., 2003*).

Interestingly, when 1 µM PUM was added to MtbRNAP, a higher percentage of RNAPs (16%) exhibited conversions to slow and super-slow inhibited states reminiscent of D-IX216 binding (*Figure 7a*). However, unlike with D-IX216, we did not observe any interconversions between the slow and super-slow inhibited states with PUM. The DTD analysis indicated that the slow inhibited states induced by PUM were composed of nucleotide addition states that were slower than wild-type (*Figure 7b*). Comparing the addition states from the slow inhibited states induced by D-IX216 and PUM, we

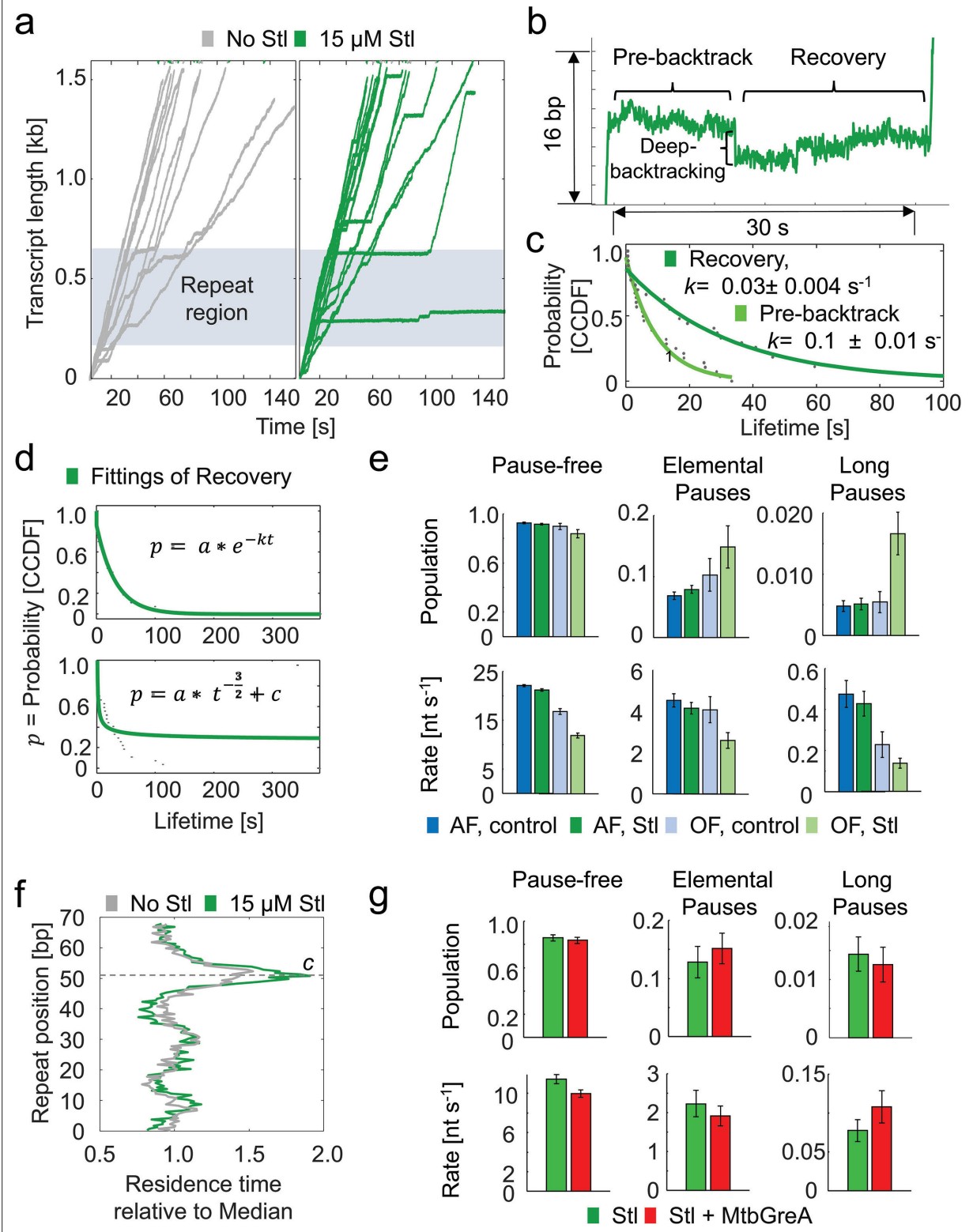

**Figure 5.** Streptolydigin inhibits *Mycobacterium tuberculosis* RNA polymerase (MtbRNAP) by enhancing pausing and inducing backtracking during pausing. (**a**) Example traces of MtbRNAP transcription in the absence (gray) and presence (green) of 15 μM Stl, with 18 pN of assisting constant force and saturating rNTP concentrations. (**b**) Details of a backtracking event induced by Stl are shown, exhibiting a process where there is both an initial pre-backtrack pause, a backtrack, and then recovery from that backtrack. (**c**) Fittings for the complementary cumulative distribution functions (CCDFs) of the pre-backtrack and recovery times of Stl-induced pauses to single exponentials are shown. (**d**) The comparison between the fitting of the

*Figure 5 continued on next page*

Figure 5 continued

backtrack recovery between two models is shown: a single-exponential distribution corresponding to deep backtracking and a power law distribution corresponding to shallow backtracking. (**e**) The kinetic parameters obtained from dwell time distribution (DTD) analysis in the presence of Stl are shown, under a constant 18 pN assisting or 4 pN opposing constant forces. Error bars are 95% CIs of fits of steps obtained from (left to right) 19, 22, 14, and 17 molecules. (**f**) Average residence time histograms of MtbRNAP transcribing the repeat of the molecular ruler in the absence and presence of Stl is shown. (**g**) Kinetic parameters extracted from DTD analysis of MtbRNAP traces under constant 5 pN of opposing force and 15 μM Stl in the absence or presence of 0.5 μM MtbGreA are shown. Error bars are 95% CIs of fits of steps obtained from 17 (-MtbGreA) and 9 (+MtbGreA) molecules.

The online version of this article includes the following figure supplement(s) for figure 5:

**Figure supplement 1.** Streptolydigin (Stl) induces reduced post-translocated states in *Mycobacterium tuberculosis* RNA polymerase (MtbRNAP).

found differences in both global velocity and addition states, similarly so for the super-slow inhibited states (*Figure 4d*, *Figure 7—figure supplement 1a, b*). Furthermore, in the presence of PUM, the polymerase displayed erratic behaviors, including the enzyme 'hopping' back and forth in 10–20 bp increments, moving steadily backward (which differs from traditional backtracking), and pausing for extended periods, sometimes for hundreds of seconds (*Figure 7—figure supplement 1c, d*). PUM had minimal effect on the RTH of the repeats, exhibiting only a slight increase in the strength of pause *c*, which we attribute to the inhibitor's preference for binding to already paused polymerases (*Figure 7c*).

## Mixing of Stl and D-IX216 shows a synergistic effect on transcription arrest

Treatment of tuberculosis usually involves a combination of antibiotics (*Harshey and Ramakrishnan, 1977*). Accordingly, we monitored transcription by MtbRNAP in the presence of both Stl and D-IX216. Notably, Stl and D-IX216 both bind to the bridge helix of the polymerase, in distinct pockets (*Lin et al., 2017*; *Temiakov et al., 2005*; *Tuske et al., 2005*; *Feng et al., 2015*), which suggests that they may affect each other's binding.

When 560 nM of D-IX216 and 15 μM of Stl were added to a transcribing MtbRNAP, the enzyme displayed either long pauses or a terminal arrest following a period of slow transcription (*Figure 8a and b*). The average duration of these long pauses, defined as those lasting over 5 s, was 25.2±5.1 s (s.e.m.) in the presence of both inhibitors. This duration is consistent with the distribution of pause durations of MtbRNAP with 15 μM Stl alone (27.5±6.7 s, s.e.m.). This suggests that these long pauses may be attributed to Stl.

Interestingly, the arrest events following periods of slow transcription were not observed with either Stl or D-IX216 alone. The velocities during these slow sections appear to correspond to the slow and super-slow inhibited states induced by D-IX216 alone (*Figure 8b*). These studies appear consistent with the hypothesis that these slow transcription periods reflect the binding of D-IX216, which transitions the RNAP into the slow or super-slow inhibited state. Subsequently, the binding of Stl leads to a permanent pause (arrest).

Interestingly, while on average it takes 60 s for an Stl molecule to bind to an actively transcribing MtbRNAP at the concentration of the inhibitor used, in the presence of D-IX216, pauses arise within ~10 s after transcription slows down. This suggests that Stl binds more readily to D-IX216-bound MtbRNAP (indicating an increased $k_{on}$) than to the transcribing enzyme alone. This observation aligns with previous findings that Stl prefers to bind to already paused polymerases (*Temiakov et al., 2005*).

In conclusion, our evidence indicates that the binding of D-IX216 can increase the $k_{on}$ of Stl, leading to the arrest of the enzyme. This suggests a synergistic effect between these two inhibitors, converting the lethargic nucleotide addition activity observed with D-IX216 alone into complete transcription arrests.

## Discussion

Our findings indicate that the global transcription velocity of MtbRNAP is slower than that of EcoRNAP, which mirrors the trend in total transcription rates observed in vivo (*Hassan et al., 2015*). In addition, whereas MtbRNAP has comparable mechanical robustness to EcoRNAP, it generally pauses more

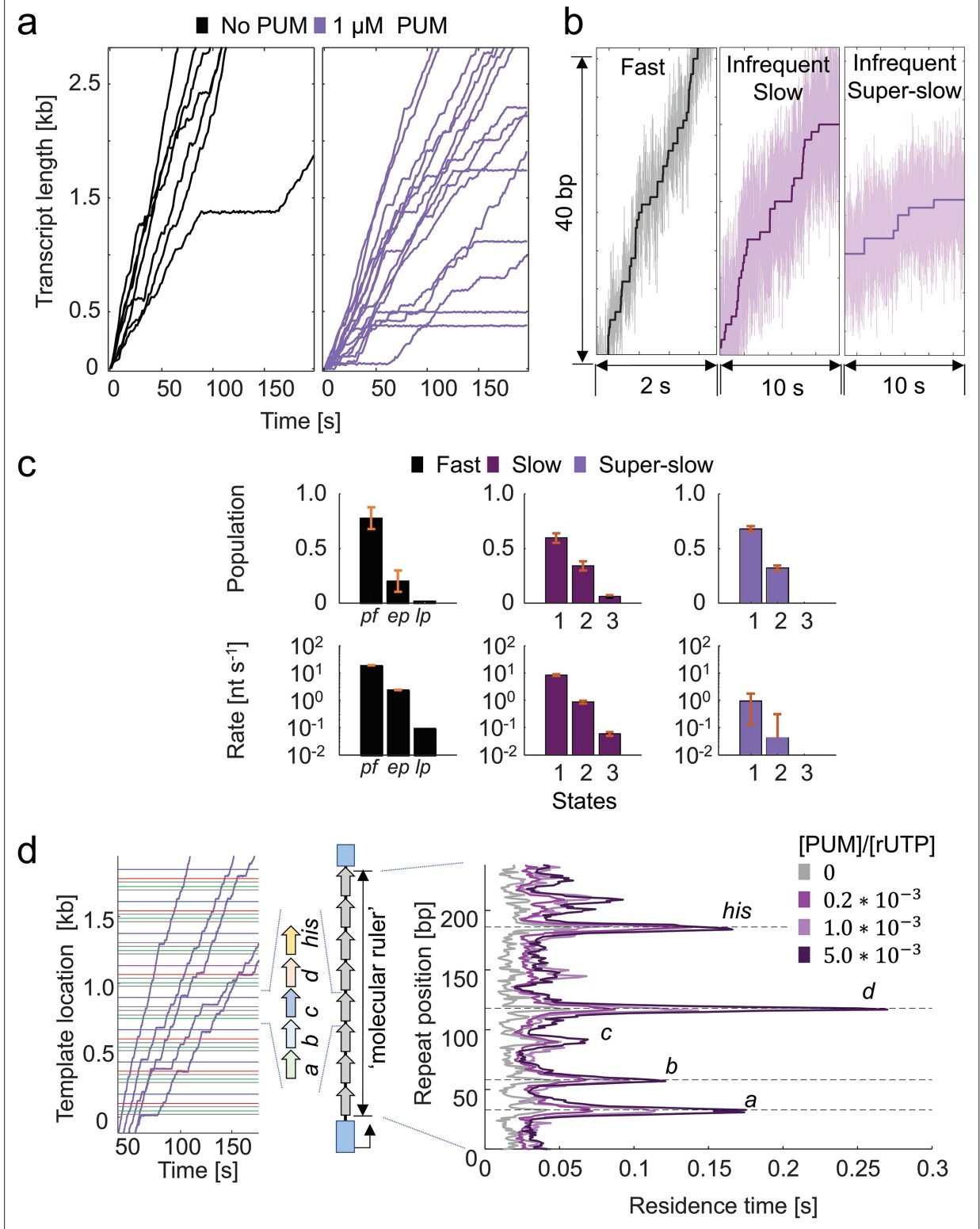

**Figure 6.** Pseudouridimycin (PUM) inhibits *Escherichia coli* RNA polymerase (EcoRNAP) by enhancing pausing and infrequently inducing slowly elongating inhibited states. (**a**) Single-molecule traces of EcoRNAP transcription in the absence (black) and presence (purple) of 1 μM PUM are shown. Traces were collected at 10 pN constant assisting force and 1 mM rNTPs. (**b**) A comparison of the fast state and the infrequent slow and super-slow inhibited states with PUM are shown. (**c**) Kinetics parameters attained from dwell time distribution (DTD) analysis of the three regions of EcoRNAP with PUM are shown. Pf: pause-free, ep: elemental pause, lp: long pause. Error bars are 95% CIs of fits of steps obtained from 27 molecules. (**d**) (left) Example

*Figure 6 continued*

traces of EcoRNAP crossing the molecular ruler in the presence of PUM are shown. (right) The average residence time histogram of EcoRNAP crossing one repeat is shown at varying ratios of PUM to 1 mM rUTP.

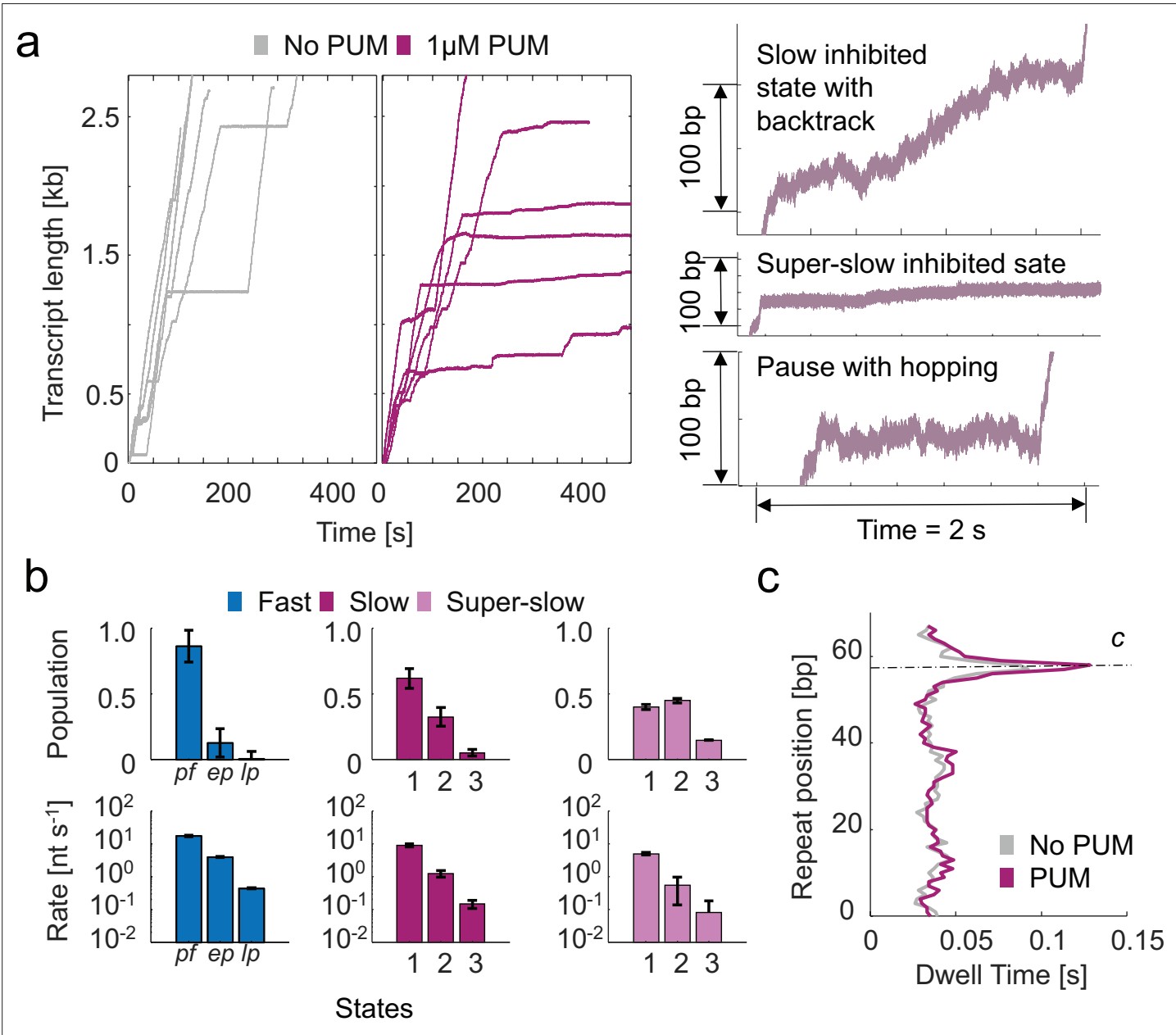

**Figure 7.** Inhibition of *Mycobacterium tuberculosis* RNA polymerase (MtbRNAP) by pseudouridimycin (PUM) involves induction of slowly elongating inhibited states. (**a**) (left) Single-molecule MtbRNAP transcription in the absence (gray) and presence (purple) of 1 μM PUM are shown. (right) Examples of transcription in the slow inhibited state, the super-slow inhibited state, and a dynamic pause are shown. Transcription was carried out under 18 pN of assisting constant force and 1 mM rNTPs. (**b**) Kinetic parameters attained from dwell time distribution (DTD) analysis of the three regions of MtbRNAP with PUM are shown. Pf: pause-free, ep: elemental pause, lp: long pause. Error bars are 95% CIs of fits of steps obtained from 27 molecules. (**c**) The average residence time histogram of a repeat of MtbRNAP transcribing the Mtb molecular ruler in the presence and absence of 1 μM PUM is shown.

The online version of this article includes the following figure supplement(s) for figure 7:

**Figure supplement 1.** Comparison of events found in transcription with antibiotics.

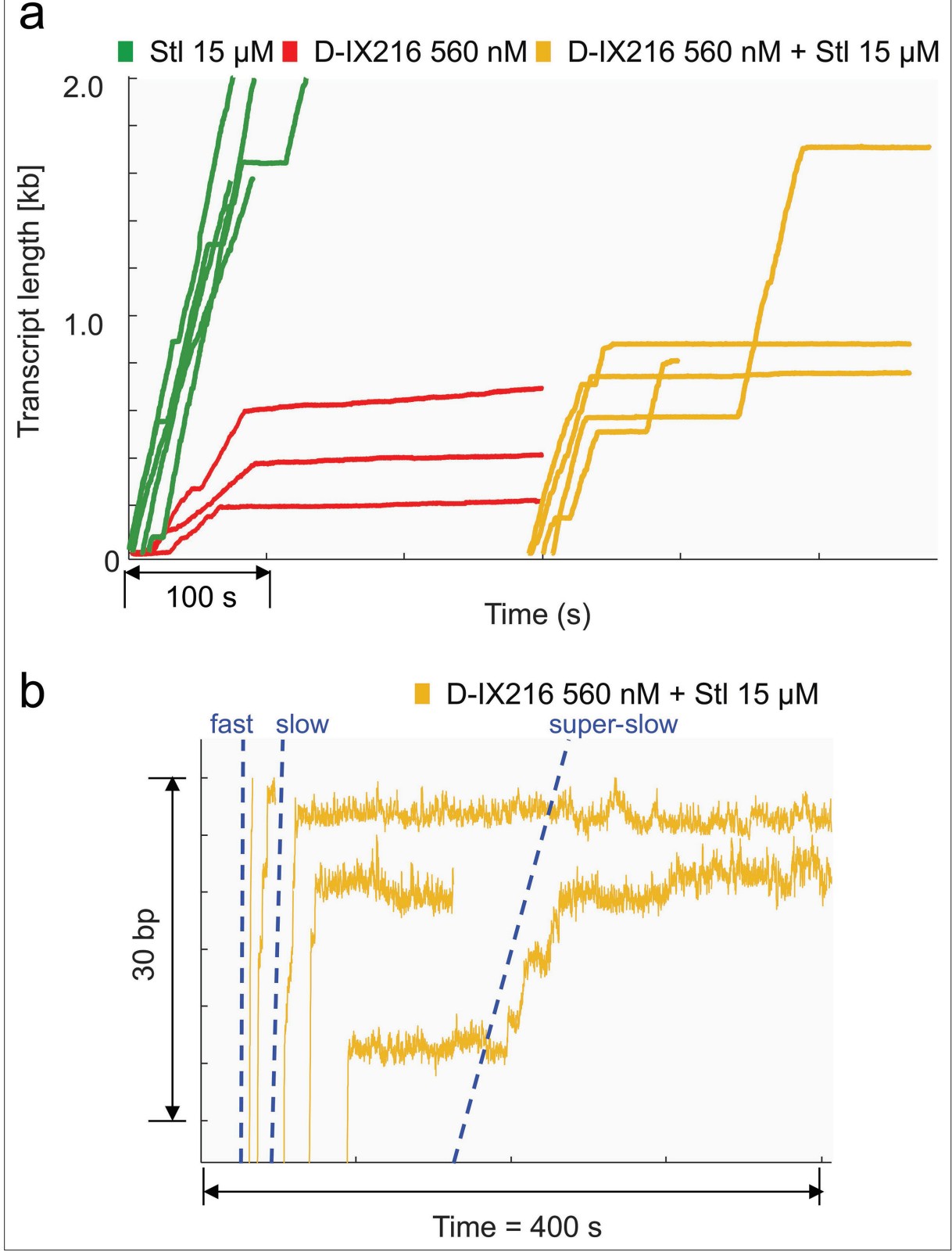

**Figure 8.** Inhibition of *Mycobacterium tuberculosis* RNA polymerase (MtbRNAP) by a combination of D-IX216 and streptolydigin (Stl). (**a**) Comparison of transcription elongation traces of single molecules of MtbRNAP obtained with D-IX216, Stl, or both are shown. (**b**) A zoom of the traces with D-IX216 and Stl just before arrest are shown. The blue dotted guide lines match the average velocity of MtbRNAP in the fast, slow, and super-slow inhibited states from left to right.

weakly or less. Our molecular ruler and bulk data suggest that the consensus pause sequences of the *E. coli* enzyme are also recognized by MtbRNAP, but again with less strength.

Our single-molecule studies demonstrate that three small-molecule inhibitors of MtbRNAP: D-IX216, Stl, and PUM do not act as chain terminators, but instead slow transcription elongation in distinct ways.

D-IX216 inhibits transcription by inducing two distinct inhibited states that incorporate nucleotides more slowly. Our findings show that this inhibitor does not completely stop nucleotide incorporation, suggesting that it is engaged during elongation and exerts an inhibitory effect that differs from other inhibitors that target the bridge helix of the RNAP (*Lin et al., 2017*; *Tuske et al., 2005*; *Feng et al., 2015*; *Bae et al., 2015*; *Malinen et al., 2015*; *Mazumder et al., 2020*; *Nedialkov et al., 2013*; *Miropolskaya et al., 2009*; *Cokol et al., 2017*). The periods of recovery from transcription elongation inhibition observed in our molecular trajectories likely correspond to inhibitor dissociation. The different states (obtained from DTD), associated with different transcription rates, may reflect stochastic differences in the functional interactions between the catalytic elements of the RNAP and the inhibitor. Future research should investigate the causes of the nucleotide addition states and inhibited states.

Previous single-molecule studies have shown that Stl lengthens existing pauses in EcoRNAP, which is consistent with our observation that pause *c* is extended with Stl in MtbRNAP. However, in those studies, EcoRNAP was not found to backtrack after inhibition by Stl, which is an interesting difference in inhibition mechanism between these two polymerases (*Arseniev et al., 2023*).

PUM increases the pause density in both EcoRNAP and MtbRNAP by competing with UTP incorporation (*Maffioli et al., 2017*). In addition, PUM can transform the active polymerase into two distinct slowly elongating inhibited states that do not appear to interconvert. We find this result surprising, as many nucleotide analog inhibitors of polymerases just cause pausing or termination, as is the case with several inhibitors being studied for use against SARS-CoV-2 (*Seifert et al., 2021*; *Dulin et al., 2015*; *Dulin et al., 2017*). It is possible that the inhibited states induced by D-IX216 and PUM share similar underlying mechanisms of inhibition, but if this is the case, it is not reflected in the different effects that these inhibitors have on the rate of elongation and DTD.

Both Stl and D-IX216 bind to the bridge helix of MtbRNAP. When administered together, D-IX216 binding seems to promote the binding of Stl, resulting in more potent inhibition (or arrest) of the enzyme. Since the standard of care for the treatment of tuberculosis infection involves a combination of antibiotics, identifying inhibitors that work together effectively is particularly valuable, and is not routinely the case (*Herrera-Asmat et al., 2017*). The results presented here with the combination of Stl and D-IX216 provide a basis for their possible use in combination therapy.

Single-molecule studies can provide details in the mechanism of action of inhibitors (*Seifert et al., 2021*; *Dulin et al., 2015*; *Dulin et al., 2017*; *Halma et al., 2023*). For instance, this study shows that PUM, ostensibly an rUTP analog, surprisingly also can induce the RNAP into states with slow activity. In addition, we see that Stl induces the RNAP to backtrack, which is a state often difficult to detect with bulk assays (*Dulin et al., 2015*). Having these additional mechanistic insights will allow medicinal chemists to make more informed choices in developing ideal combinations of drugs against multidrug-resistant diseases such as tuberculosis.

## Methods

A list of oligos, plasmids, and cells used are available in *Supplementary file 1*. A list of conditions studied in the optical tweezers assay, including numbers of traces for each, is included in *Supplementary file 2*.

### Plasmid preparation

To make the vectors for MtbRNAP biotinylation, we performed round-the-horn site-directed mutagenesis of plasmid pET-DUET-BC *Gradia et al., 2017* using long PCR to introduce an avitag or a sortase tag (sortag) sequence at the c-terminus of the rpoC gene (*Gabizon et al., 2018*). The resulting plasmids were named pET-DUET-BC-avitag or pET-DUET-BC-sortag, respectively. As for the vectors for expressing Mtb transcription factors, we amplified the corresponding gene from the Mtb genome

and inserted it into the plasmid pET His6 TEV LIC (Addgene, # 29653) using ligation-independent cloning (*Chen et al., 2011*).

To make plasmids with the DNA templates for the single-molecule assays, we took pUC19 and cloned the promoter AC50, the 10 bp stalling cassette, and the transcribable region, which derived from a ~2.7 kb region of genes rpoB and C from the *M. tuberculosis* Erdman genome (a gift from Professor Sara Stanley). We used an additional round-the-horn site mutagenesis to remove the lac promoter. We named this plasmid pUC19-del-Lac-AC50-MtbrpoBC. We used this plasmid to create another plasmid containing a molecular ruler. First, we removed one side of the BsaI in the rpoC gene by inserting a 255 bp DNA fragment of terminator WhiB1. Second, we replaced the stalling sequence with a 20 bp CU-less cassette using round-the-horn mutagenesis. Third, we used NEBuilder HiFi DNA Assembly Master Mix (NEB E2621S) to insert a synthetic DNA fragment, an 8 x repeat of a 63 bp sequence of pause *c* (Genescript), into the rpoB gene. The resulting plasmid was named pSGUML1T1.

## DNA template preparation

We made a single-end digoxygenin-labeled DNA template (without a ruler) using PCR by amplifying approximately 4.2 kb or 2.8 kb from plasmid pUC19-del-Lac-AC50-MtbrpoBC, depending on whether the assay is for opposing or assisting force, and utilized a dig-labeled primer.

To make the DNA template containing the molecular ruler, we linearized the plasmid pSGUML1T1 with BsaI and filled the sticky ends by using Klenow Fragment in the presence of either ddATP or ddCTP to produce the opposing or assisting force template, respectively (*Gabizon et al., 2018*).

In addition, DNA handles were created using PCR with a biotinylated primer along with another primer, either double-dig labeled or encoding a restriction enzyme site for generating sticky ends for later ligation.

## Production of biotinylated and non-biotinylated MtbRNAP

We produced the holoenzyme of MtbRNAP with sigma factor A by co-expressing its subunits in *E. coli* (*Gradia et al., 2017*). We biotinylated the avitag-MtbRNAP by co-expressing the plasmids pACYD-DUET-AOS and pDUET-BC-avitag with the plasmid pCDF-BirA (a gift from Professor Nicola Burgess-Brown). This protein was used for the assay under opposing force with StI and GreA. Alternatively, we biotinylated the sortag-MtbRNAP (LPETG at the c-terminal of β') after purification using an in vitro reaction of the polymerase, the enzyme sortase A (*Chen et al., 2011*) and GGGL-biotin (Genscript) (*Righini et al., 2018*).

## Production of transcription factors MtbCarD and MtbGreA

We purified the transcription factor CarD as before (*Gradia et al., 2017*). To obtain MtbGreA, we expressed it as a his-tagged protein in *E. coli* BL21(DE). We captured this protein with Histrap (Cytiva) and used a TEV protease cleavage reaction (*Sidorenkov et al., 1998*) to cleave its histidine tag. To remove the cleaved tag, we used reverse nickel chromatography. The final purification step was done with anion exchange chromatography (HiTrap Q, Cytiva) in a Tris-saline buffer.

## In vitro transcription bulk assay

We used an artificial bubble-based transcription to assess the recognition of the elemental pause sequences by MtbRNAP using a FAM-labeled RNA (*Gradia et al., 2017*) and assessed transcript size via gel. Raw gels are available in the Source Data Files related to the relevant figures.

## Ternary elongation complex formation

We stalled the biotinylated MtbRNAP on a cassette lacking one or two nucleotides to form the ternary elongation complex (TEC) to study elongation, as previously described (*Nudler et al., 2003*; *Berg-Sørensen and Flyvbjerg, 2004*).

## Sample preparation for single-molecule optical tweezers transcription elongation assay

To create the promoter-initiated stalled MtbRNAP elongation complexes, we utilized MtbCarD. First, we pre-incubated approximately 1 μM of biotinylated-MtbRNAP with around 10 μM of MtbCarD at 37 °C for 5 min in a final volume of 2 μL. Subsequently, we diluted the biotinylated MtbRNAP in buffer

TB40 (20 mM Tris-HCl pH 7.9, 40 mM KCl, 5 mM MgCl$_2$, and 0.5 mM TCEP) to a concentration of 10 nM. We then stalled 5 nM of polymerases by incubating them with 150 µM GpG, 2 nM Mtb DNA template in stall buffer (20 mM Tris-HCl pH 7.9, 3 µM rATP, 3 µM rGTP, and 3 µM rCTP) at 37 °C for 20 min. If we were to use the Eco DNA repeat, we would substitute GpG with ApU (*Whitley et al., 2017*).

To attach the DNA handles or stall complexes to the microbead surfaces, we used a few different methods. In the first method, we incubated a 0.1 nM stalled complex labeled with double-digoxygenin with 0.05–0.1% passivated anti-dig beads in TB40 buffer for 10 min at 22 °C. Then, we added 0.05 µg/µL heparin at 22 °C for 5 min to remove non-competent polymerases. In the second method, we incubated 0.2–0.5 nM digoxygenin-labeled DNA handle with 300 nM neutravidin and the same amount of anti-digoxigenin beads at 22 °C for 10 min, followed by the addition of heparin. In the third method, we conducted ligation-based deposition by incubating a 0.1 nM stall complex with 0.1–0.05% passivated oligo beads, 0.1 mM rATP, and 4 U/µL T4 DNA ligase in TB40 buffer at 22 °C for 45 min to facilitate the ligation reaction. This was followed by a 5 min treatment with heparin at 22 °C. In the fourth method, we attached 0.2–0.5 nM DNA handle by incubating it with passivated complementary oligo beads, 300 nM neutravidin, 0.1 mM rATP, and 4 U/µL of T4 DNA ligase in TB40 buffer for an identical prolonged incubation, followed by heparin treatment. The bead depositions were either kept on ice for immediate use, or preserved by flash freezing and stored at –80 °C.

For every 10 µL of deposited beads, we added 1000–1500 µL of degassed TB130 Buffer (20 mM Tris-HCl pH 7.9, 130 mM KCl, 10 mM MgCl$_2$, 1 mM TCEP, and 5 mM NaN$_3$) before injecting into the fluidics chamber.

In the experiments with MtbCarD and MtbGreA, the final amount of initiating CarD contributed to the amount free in solution in elongation was as low as ~13 pM (diluted from 100 to 120 nM MtbCarD), and neither causes any effects on pausing or velocity.

## High-resolution optical tweezer transcription elongation assay

In *Figure 1a*, there is a representation of the optical tweezers setup, and in *Figures 1b, c and 2b*, there are representations of assisting constant force, opposing constant force, and opposing passive mode experiments. We used a custom home-built timeshared dual-trap optical tweezers system (*Whitley et al., 2017*; *Comstock et al., 2011*). In short, the single-molecule assay to measure transcription elongation involves two beads, one coated with RNAP stalled complexes, and one coated with DNA handles with neutravidin on the free end. The two beads are caught in separate traps, and when the beads are brought in close proximity, the neutravidin on the end of the DNA handle binds to the biotin attached to the RNAP and a tether is formed. The beads are brought apart, and the DNA handle separates the RNAP from the surface of the bead to reduce photodamage. We also used an oxygen-scavenger system based on sodium azide to increase the robustness of collecting processive activity. For both EcoRNAP and MtbRNAP, the attachment to biotin is done via its β' subunit.

We injected the microbeads into a microfluidics chamber through different channels to diffuse them to a central main channel, trapped them by optical tweezers, and brought them close to each other from the tether. To resume the elongation activity, we flowed a solution containing rNTPs, transcription factors, and/or inhibitors through a shunt positioned very close to the tethered complex. We used fluidic valves to control the flow via gravity. When transcribing the Eco repeat sequence, we used a nucleotide cocktail of 1 mM rUTP, 1 mM rGTP, 0.5 mM rATP, and 0.25 mM rCTP *Gabizon et al., 2018* to match previous studies.

The instrument records the trap separation and monitors the displacement of the beads from the centers of the traps, and handles force feedback for the constant force experiments (*Tropea et al., 2009*). Calibration from raw voltages from the instrument to distance (bead separation) and applied force is done via power spectrum calibration to the expected Brownian motion of a bead held in a trap (*Whitley et al., 2017*).

## Analysis of single-molecule complete trajectories of transcription elongation

All analysis, including the processing of the raw instrument data to forces and bead-to-bead distances and analysis procedures is done via custom code written in MATLAB, and is available at https://doi.org/10.5281/zenodo.15232932.

The collected traces are first selected for their length and undergo an initial sorting step where traces with unusually high instrumental noise or those that are unfit for analysis (two RNAPs tethered at the same time, too short transcription, etc.) are removed.

For calculating velocity distributions, a Savitzky-Golay differentiating filter of degree 1, width 1 s is used to extract the velocity of the trace at every point. These velocities are binned and least-squares fit to the sum of two Gaussians, one with mean 0 bp/s.

To get the dwells of the polymerase, we fit the traces as a staircase with a 1 bp step size. The complementary cumulative distribution function (CCDF) of the dwell time distribution (DTD) is fit to a sum of exponentials $\sum_{i=1}^{n} a_i e^{-k_i t}$, where $i$ represents the $i$-th nucleotide addition state of the RNA polymerase, $a_i$ is the state probability ($\sum_{i=1} a_i = 1$), and $k_i$ is the kinetic rate such that $k_1 > k_2 > \dots k_n$. The CCDF is fit to the sum of up to 6 exponentials, and the best is chosen via the Akaike information criterion.

We calculated the residence time histogram (RTH) of the polymerase transcribing the pauses repeat according to a previous publication (*Gabizon et al., 2018*). Briefly, the repeating pauses let us determine a scale factor and offset factor to each transcription trace based on the location of the pauses. The scale is chosen to space the pauses 68 bp apart, and the offset is chosen to place the pause at position 50 bp into the repeat. This is done by taking the RTH of the repeat, averaged across the 8 repetitions, for a range of scale factors. The one that aligns the pauses best is the one with the most peaked RTH (a poor scale factor does not align the pauses, so makes the RTH flat, a good scale factor aligns the pauses and thus it will show up as a peak). The offset factor is then chosen to place the location of the strongest peak in the best RTH at 50 bp.

For the analysis of passive mode data, individual stalling force-velocity profiles are generated and the force $F$ and velocity $V$ is transformed into normalized units $v$ and $f$, with the velocity normalized by dividing by the maximum force $V_{max}$ and the force normalized by dividing by the stall force, which is the force at which the velocity is half of $V_{max}$. These individual force-velocity profiles are averaged together and fit to the function $v(f)=(1+A)/(1+Aexp(f\delta/kT))$. For a short derivation, let the nucleotide addition cycle be made up of two steps that occur sequentially, one force-independent (comprising the biochemical transitions) and force-dependent (comprising the mechanical transitions). Let the rates of these terms be $k_{biochemical} = k_b$ and $k_{mechanical}(F)=k_m*exp(-Fd/kT)$. For the mechanical term, the system must move some distance $d$ against the optical tweezers' force $F$ to reach the transition state, so the transition state energy is raised by $Fd$ and the rate is affected via a Boltzmann factor. The transcription velocity is then the inverse of the sum of the inverses of these two rates: $V(F) = (k_b^{-1}+k_m^{-1}exp(Fd/kT))^{-1}$. Let $V_{max} = V(0)=1/(k_b^{-1}+k_m^{-1})$, so the normalized velocity $v(F)=V(F)/V_{max} = k_b^{-1}+k_m^{-1} /(k_b^{-1}+k_m^{-1}exp(Fd/kT))$. Let $A=k_b/k_m$ and let $Fd = f\delta$ (to transform $Fd$ (unit pN * nm) into $f\delta$ (unit normalized force * 'normalized distance')) to get the stated relation, $v(f)=(1+A)/(1+Aexp(f\delta/kT))$. A is a parameter that dictates the ratio between the biochemical (force-independent) and mechanical (force-dependent) rates, and $\delta$ is the force dependence of the mechanical term. We divide 'normalized distance' by $F_{stall}$ to get the distance to transition state in length units since $Fd = f\delta$ and $f=F/F_{stall}$.

## Acknowledgements

We would like to extend thanks to Keren Espinoza and Cristhian Cañari for their work in the early stages of this project, and Drs. S Maffioli and S Donadio of Naicons Srl for samples of PUM used in initial work. This research was funded by National Institutes of Health grants GM041376 to RHE and R01GM032543 to CB. WL was supported by the National Natural Science Foundation of China grant 31900883, Department of Biomedical Engineering, Shenzhen University. CB is a Howard Hughes Medical Institute investigator.

## Additional information

### Funding

| Funder | Grant reference number | Author |
|---|---|---|
| National Institutes of Health | GM041376 | Richard H Ebright |
| National Institutes of Health | R01GM032543 | Carlos Bustamante |
| National Natural Science Foundation of China | 31900883 | Wenxia Lin |
| Howard Hughes Medical Institute | | Carlos Bustamante |

The funders had no role in study design, data collection and interpretation, or the decision to submit the work for publication.

### Author contributions

Omar Herrera-Asmat, Conceptualization, Resources, Software, Formal analysis, Validation, Investigation, Visualization, Methodology, Writing – original draft, Project administration, Writing – review and editing; Alexander B Tong, Resources, Data curation, Software, Formal analysis, Visualization, Writing – review and editing; Wenxia Lin, Validation, Investigation, Methodology, Writing – review and editing; Tiantian Kong, Juan R Del Valle, Carlos Bustamante, Conceptualization, Resources, Supervision, Funding acquisition, Writing – review and editing; Daniel G Guerra, Conceptualization, Resources, Writing – review and editing; Yon W Ebright, Richard H Ebright, Conceptualization, Writing – review and editing

### Author ORCIDs

Omar Herrera-Asmat ⬤ https://orcid.org/0000-0002-7965-444X
Alexander B Tong ⬤ https://orcid.org/0000-0003-3793-2533
Juan R Del Valle ⬤ https://orcid.org/0000-0002-8315-5264
Daniel G Guerra ⬤ https://orcid.org/0000-0002-2410-722X
Richard H Ebright ⬤ https://orcid.org/0000-0001-8915-7140
Carlos Bustamante ⬤ https://orcid.org/0000-0002-2970-0073

### Decision letter and Author response

Decision letter https://doi.org/10.7554/eLife.105545.sa1
Author response https://doi.org/10.7554/eLife.105545.sa2

## Additional files

### Supplementary files

Supplementary file 1. Tables of oligos, plasmids and cells.

Supplementary file 2. Summary of traces.

MDAR checklist

### Data availability

All oligos/plasmids used in this study are available upon reasonable request. See *Supplementary file 1* for a list of materials. Transcription traces are available from Dryad. Matlab code used for analysis is available on Zenodo.

The following dataset was generated:

| Author(s) | Year | Dataset title | Dataset URL | Database and Identifier |
|---|---|---|---|---|
| Herrera-Asmat O, Tong AB, Bustamante C | 2025 | Single-molecule optical tweezers data of transcription of *M. tuberculosis* and *E. coli* RNAP in the presence of single-molecule inhibitors | https://doi.org/10.5061/dryad.2fqz6130m | Dryad Digital Repository, 10.5061/dryad.2fqz6130m |

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
