## [Editor Report]

This important work used state-of-the-art optical tweezers to investigate the elongation of *Mycobacterium tuberculosis* RNAP (mtbRNAP) and its inhibition by three small-molecule inhibitors: N(α)-aroyl-N-aryl-phenylalaninamide (D-IX216), streptolydigin (Stl), and pseudouridimycin (PUM). The authors provide compelling evidence demonstrating a slower elongation mode of MtbRNAP compared to that of the *E. coli* RNAP, distinct inhibition modes of the three small molecule inhibitors, and a synergistic effect of Stl and D-IX216 on MtbRNAP. These findings clarified the differential mechanistic actions of RNA polymerase inhibitors, providing insights for the development of new, effective combination therapies against tuberculosis.

---

## [Decision Letter]

**Decision letter after peer review:**

Thank you for submitting your article "Pleomorphic effects of three small-molecule inhibitors on transcription elongation by *Mycobacterium tuberculosis* RNA polymerase" for consideration by *eLife*. Your article has been reviewed by 2 peer reviewers, and the evaluation has been overseen by a Reviewing Editor and Volker Dötsch as the Senior Editor.

Essential Revisions:

1) please provide each measurement's N value (number) of data points to allow statistical significance. When stating "significant," please explicitly describe the statistical test or analysis method used.

2) Please clarify inconsistent nomenclatures as indicated by reviewer #1.

3) Please clarify or repeat the same concentrations of D-IX216 in the individual vs. combined inhibitor treatments with Stl.

*Reviewer #1 (Recommendations for the authors):*

The studies by Herrera-Asmat et al., appear to be carefully performed and the statistical analysis of the data appropriate. However, there are a number of nomenclature inconsistencies and several unsupported conclusions as well as missing primary data that are both necessary and must be addressed before publication. These additions will significantly improve presentation of the data and conclusions for the wide audience of *eLife* and assure the experimental reproducibility that is essential to move the field forward.

There are a number of data and editorial issues that made evaluation of the Herrera-Asmat et al., manuscript quite difficult and time consuming. In general, the data appear to be carefully collected, and the statistical evaluation well thought through. Nevertheless, the authors need to address several issues to improve both the clarity and accuracy of the manuscript. These are:

1) P.3, L.92-93: nowhere in the manuscript are the number of (individual) molecules analyzed noted. The optical tweezers setup can only observe a single RNAP molecule at a time. These observations may include multiple kinetic events from individual molecules that underpin statistical significance. However, to understand the general nature of these kinetic events, one must observe and characterize multiple individual molecules to provide power to any statistical analysis.

2) P.3, L.118-125: The authors define three kinetic states of the RNAP; pause-free (pf), elemental pause (ep) and long pause (lp). However, this abbreviated nomenclature is not used consistently throughout the manuscript, making reading and evaluation difficult at best. This is complicated by random changes in this nomenclature that include "on-pathway (for pf) and short pauses (for ef). Moreover, it was not clear to this referee whether there was a connection between pf, ef, and lp to the fast, slow and super-slow activities observed for the RNAP's in the presence of drug inhibitors. This should be explicitly considered in the text.

3) P.4, L.134: Isn't the "passive mode" functionally equivalent to the "opposing force" mode? Ultimately the authors clearly determine at which "opposing" force the polymerase stalls. This seems to be another case where consistent use of nomenclature would help the reader.

4) P.4, L.150-153: This is a rather confusing sentence. By definition backtracking must occur after a stall in forward polymerase activity. Thus, δ seems more likely to be the nucleotide distance to the transition where the opposing mechanical force equals the polymerization force?

5) P.5,L.157: Isn't it more accurate that, MtbRNAP tends to pause less frequently than EcoRNAP under "both" assisting "and opposing" force "for" both the Eco and Mtb templates?

6) P.6,L.174-175: This is a rather speculative conclusion without some genetic or mutational data and should be either eliminated or dramatically tempered.

7) P.5, L.186-189: Inconsistent (often confusing) nomenclature. The authors define the drug inhibitor D-IX216 as "AAP" in the introduction. Then inconsistently use D-IX216 and/or AAP throughout the manuscript. Either refer to AAP as the class of drugs and D-IX216 as the specific drug used in the studies, or AAP as the acronym for D-IX216, but not both or either depending on the mood.

8) P.5,L.193: The force conditions for this drug concentration analysis should be explicitly detailed.

9) P.6,L228-229 (and throughout the manuscript): The use of the word "region" does not seem appropriate. This really is a measure of polymerase "activity". There would seem to be no reason not to refer to these kinetic regions as fast, slow and super-slow (polymerase) activities.

10) P.6,L.250 and 252: Figure 4e should refer to the left and right panels, respectively, to focus on specific results.

11) P.7,L.285-290: These sentences seem repetitive and are quite confusingly written. There must be a better way to express the differences in mathematical treatment of deep backtracking and shallow backtracking mechanics, and the relative fitting that describes the results.

12) P.8,L.307-308: These are not "seemingly conflicting" results. They are just "different results".

13) P.8,L.331: Is this AAP's as a general class, or specifically D-IX216? I read it as the latter, but these continued confusing nomenclatures made simple reading of the manuscript quite difficult!

14) P.9,L.350: Two different concentrations of D-IX216 were used in the individual vs. combined inhibitor treatments with Stl. This difference makes these studies not exactly comparable and may change the results and interpretation. A repeat of these studies with comparable concentrations of drug is essential for this entire section and is a significant flaw in these studies.

15) P.9,L360: Rather than, "Therefore we propose", one might consider, "These studies appear consistent with the hypothesis …".

16) P.9,L389: "unbinding" should be replaced by "inhibitor dissociation".

17) P.10,L.397-399: There must be a much simpler way of expressing the differences and relative value of single molecule studies vs. bulk studies? There is no question that the present studies identified new kinetic processes that would be impossible by bulk analysis. But the present sentence obscures this fact with repetitive words and pretentious language.

18) P.10,L.412-415: It would seem to this referee that the final sentence could better reflect the capabilities of single molecule analysis for antibiotic mechanics in developing combined treatments for improved patient outcomes?

19) Figure 2d (right panel): Equations should be included in Supplementary Materials with terms for Boltzman relation defined (not presently included anywhere in the manuscript). This could include the actual calculations, which were not immediately obvious and required significant time to deconvolute.

20) Figure 3b (right panel): Marking pause sites (a,b,c,d,his) would visually help readers.

21) Figure 3c (right panel): Marking pause site bands (a,b,c,d,his) for MtbRNAP would visually help readers. One assumes that the MtbRNAP pause site c is the major band and ultimately made up the MtbRNAP pause site array? Why was pause site b not used as an equivalent array? Seems a strong pause at least in early time points.

22) Figure 4a (left panel): What is the concentration of D-IX216?

23) Figure 4a (right panel): The pause dwell times for the fast kinetics prior to the super-slow activity seem different that the fast or slow activities?

24) Figure 5e and 5g: The authors must use consistent nomenclature for pf, ef, and lp in these panels. Moreover, the meaning of AF (assisted force) and OP (opposing force) should be included in the legend.

25) Figure 6c: The illustration of polymerase kinetic activity and State 1,2,3 is different than Figure 4d. It would be useful if the authors stuck to a type of illustration to make comparisons of Figures easy for the reader.

26) Suppl. Figure S2c: Use pf and ep, or some consistent nomenclature!

27) Suppl. Figure S4c: Same as #26 – consistent nomenclature, please!

Methods:

1) The studies were not performed at physiological ionic strength (looks like ~70mN). One wonders if there might be significant kinetic differences.*Reviewer #2 (Recommendations for the authors):*

Referee report for "Pleomorphic effects of three small-molecule inhibitors on transcription elongation by *Mycobacterium tuberculosis* RNA polymerase" by Omar Herrera-Asmat et al.

In this work, the authors use their well-established single-molecule optical tweezers assay to study the activity of the RNA polymerase from *Mycobacterium tuberculosis*. In particular, they investigate the changes in activity in the presence of three small-molecule inhibitors: N(α)-aroyl-N-aryl-phenylalaninamide (D-IX216), streptolydigin (Stl), and pseudouridimycin (PUM). They find a diverse range of effects on the polymerase, including transitions into a very slow activity state (induced by D-IX216 and PUM), induction of pausing and backtracking. Overall, the work provides interesting insights into the activity of an important pathogen with implications for the development of new drugs. This is a timely and relevant study that I found well-presented and interesting to read. Below, are some suggestions for clarifications and further improvements.

- In several places they authors make statements like "there are no significant effects" (line 133) or "we observed an insignificant decrease" (line 307). It would be good to explicitly state what statistical test / method of analysis was used.

- Related to the last point: In many of the figures there are bar charts essentially comparing various kinetic parameters. I believe it would be useful to -at least for key parameters- indicate which differences are statistically significant and which ones are not.

- The model used in Figure 2 to describe the force-dependent transcription velocity is not well explained. There is "a parameter A" (line 146), but then there is also lower case "a", which is never defined? Also, the authors give values for the stall force, which does not appear to be a parameter in the model, but do not seem to give values for Vmax, which is used to rescale the axes. I think the model should be better explained and all parameters defined. In addition, I am wondering whether it would be clearer and more informative to show velocity vs. force for the two polymerases, rather than to normalize both axis, which makes the plots look very similar.

- I believe it would be useful to (further) compare and contrast the present findings with previously published work on how different inhibitors affect RNA polymerases, see e.g. single-molecule work on SARS-CoV-2 (Seifert et al. *eLife* 2021; https://doi.org/10.7554/*eLife*.70968).

---

## [Author Response]

Essential Revisions:Reviewer #1 (Recommendations for the authors):The studies by Herrera-Asmat et al., appear to be carefully performed and the statistical analysis of the data appropriate. However, there are a number of nomenclature inconsistencies and several unsupported conclusions as well as missing primary data that are both necessary and must be addressed before publication. These additions will significantly improve presentation of the data and conclusions for the wide audience of eLife and assure the experimental reproducibility that is essential to move the field forward.There are a number of data and editorial issues that made evaluation of the Herrera-Asmat et al., manuscript quite difficult and time consuming. In general, the data appear to be carefully collected, and the statistical evaluation well thought through. Nevertheless, the authors need to address several issues to improve both the clarity and accuracy of the manuscript. These are:1) P.3, L.92-93: nowhere in the manuscript are the number of (individual) molecules analyzed noted. The optical tweezers setup can only observe a single RNAP molecule at a time. These observations may include multiple kinetic events from individual molecules that underpin statistical significance. However, to understand the general nature of these kinetic events, one must observe and characterize multiple individual molecules to provide power to any statistical analysis.

We thank the reviewer for raising this point. We have now tabulated the number of traces (=number of individual RNAPs) from each condition tested in this work in Supplementary File 2, and have added references to the text and methods denoting that N information can be found there. For instance, one reference in Methods reads: “A list of conditions studied in the optical tweezers assay, including numbers of traces for each, is included in Supplementary File 2”

2) P.3, L.118-125: The authors define three kinetic states of the RNAP; pause-free (pf), elemental pause (ep) and long pause (lp). However, this abbreviated nomenclature is not used consistently throughout the manuscript, making reading and evaluation difficult at best. This is complicated by random changes in this nomenclature that include "on-pathway (for pf) and short pauses (for ef). Moreover, it was not clear to this referee whether there was a connection between pf, ef, and lp to the fast, slow and super-slow activities observed for the RNAP's in the presence of drug inhibitors. This should be explicitly considered in the text.

The reviser raises a valid point. We now have unified nomenclature for these states and have explicitly defined them. To be clear, the analysis of the dwell time distributions yields kinetic states k_1_, k_2_, …. For the inhibitor-free polymerase, we relate these k_1_, k_2_, and k_3_ to k_pf_, k_ep_, and k_lp_, respectively, since the kinetic lifetimes of the k_1/2/3_ states match the lifetimes of the k_pf/ep/lp_ states that have been previously described in other studies (e.g. https://doi.org/10.1016/j.celrep.2022.110749). For the states observed under the action of D-IX216 (slow and super-slow), the analysis of the dwell time distributions also yields a k_1_, k_2_, and k_3_, but we do not relate those to k_pf_, k_ef_, and k_lp_. We have made this point clearer in the main text when this is discussed.

The labeling of states in Figures 1e, 2a, 4d, 5e, 6c, 7b, S2c (Figure 4 supplement 1), and S4c (Figure 4 supplement 3), have been changed to Pause-free, Elemental Pauses, and Long Pauses.

When discussing the slow and super-slow DTD analysis, we emphasize the differences between the states observed in the presence of D-IX216 and pf/ep/lp, relevant section copied here: “We emphasize here that inhibited states refer to a persistent activity of the enzyme lasting for multiple incorporation cycles, while nucleotide addition states may change between every step. The rates of the various nucleotide addition states found for the slow and super-slow inhibited states do not match those of the pause-free, elemental pause, or long pause states (Figure 4d), so we conclude that these are states distinct from the normal ones visited by the polymerase.”

3) P.4, L.134: Isn't the "passive mode" functionally equivalent to the "opposing force" mode? Ultimately the authors clearly determine at which "opposing" force the polymerase stalls. This seems to be another case where consistent use of nomenclature would help the reader.

The reviewer is correct. Passive mode is an experimental procedure where the distance between the optical traps is kept fixed and therefore, as the enzyme moves, the force opposing the molecule increases. Thus, the difference between ‘passive mode’ and ‘opposing force mode’ as currently written in the text is as to whether force feedback is applied to keep the force constant at a series of pre-set values (‘opposing force mode’), or to let the force rise (in real time) as the polymerase transcribes (‘passive mode’). We have edited the terms to be clearer: the experimental modes are now ‘opposing passive mode’, ‘opposing constant force mode’, and ‘assisting constant force mode’. The relevant portions of the text and figures have been updated.

4) P.4, L.150-153: This is a rather confusing sentence. By definition backtracking must occur after a stall in forward polymerase activity. Thus, δ seems more likely to be the nucleotide distance to the transition where the opposing mechanical force equals the polymerization force?

We agree that δ relates to the attainment of the stall state, but since that stall state is the backtracked state (as evident in visible backwards movement that occurs with stalling), we interpret it as being related to the degree of backtracking. If δ is related to the mechanical inhibition of polymerase forwards translocation (“fighting against the polymerization force”), then a value of δ close to the step size of the motor, 1bp, would have been obtained, but the experimental value of ~15bp strongly suggests that it describes a quantity related to the backward movement (backtracking) of the enzyme.

To elaborate, there is probably an intermediate stall state between forward translocation and backtracking, but since there is presumably no change in tether extension in the attainment of this state, the rate of attaining this state cannot be force-dependent. So, stalling is probably a stochastic process by which the polymerase transiently stochastically pauses, from which it can transit to a backtracking state, and the rate of going from the stochastic pause to the backtrack pause involves backward displacement and is therefore force-dependent with the opposing force promoting movement towards the backtracking direction. This process (a polymerase even in the absence of any drug first enters a short pause, from which it can enter a backtrack pause) matches proposed mechanisms for polymerase backtracking, e.g. see https://doi.org/10.1016/j.celrep.2022.110749.

We have edited the text to try to be clearer on these points and on the model itself. The main text section now reads: “The force-velocity curves can be fit to an unrestrained Boltzmann relation *v(f)=(1+A)/(1+Aexp(fδ/kT))*, where the process is split into mechanical (force-dependent) and biochemical (force-independent) terms. The ratio of these terms at zero force is the parameter A, and the strength of force dependence is *δ* (Figure 2d, see Methods). For both Eco and Mtb RNAPs, we find that A << 1, indicating that biochemical transitions are rate-limiting at zero force in these instances, in line with prior findings for EcoRNAP. The value of δ is the distance to the transition state of polymerase stalling via backtracking, so we interpret this as the backtracking distance required by the RNAP to stall. For MtbRNAP, we obtain a value of 14 ± 1 nt (95% CI) (Figure 2d), comparable to the 13 ± 1 nt observed for EcoRNAP and findings from earlier studies on EcoRNAP.” We have also added a section to Methods where this model is derived and discussed in more depth.

5) P.5,L.157: Isn't it more accurate that, MtbRNAP tends to pause less frequently than EcoRNAP under "both" assisting "and opposing" force "for" both the Eco and Mtb templates?

While the reviewer’s suggested statement would probably be true, Figure 1e does not relate to any opposing force data, and we don’t show any analysis directly comparing EcoRNAP and MtbRNAP in constant opposing force mode. So, we opt to keep the text as-is.

6) P.6,L.174-175: This is a rather speculative conclusion without some genetic or mutational data and should be either eliminated or dramatically tempered.

We agree with the reviewer on this point and have elected to remove this statement.

7) P.5, L.186-189: Inconsistent (often confusing) nomenclature. The authors define the drug inhibitor D-IX216 as "AAP" in the introduction. Then inconsistently use D-IX216 and/or AAP throughout the manuscript. Either refer to AAP as the class of drugs and D-IX216 as the specific drug used in the studies, or AAP as the acronym for D-IX216, but not both or either depending on the mood.

Our intention was to do as you suggest: to refer to the family of drugs as AAPs, while the specific one we have tested is D-IX216. The phrase in the introduction (“the N(α)-aroyl-N-aryl-phenylalaninamide (AAP) D-IX216”) is unclear, we meant to define AAP as the shortening of N(α)-aroyl-N-aryl-phenylalaninamide, separate from D-IX216. We realize we don’t have to define AAP, so we have removed this abbreviation. The sentence now reads “These inhibitors are D-IX216 (an N(α)-aroyl-N-aryl-phenylalaninamide), streptolydigin (Stl), and pseudouridimycin (PUM),” which should be clearer. We have eliminated all reference to AAP otherwise. In particular, we referred to the drug we tested as “AAP” in a few places in the text. These have been changed to D-IX216.

8) P.5,L.193: The force conditions for this drug concentration analysis should be explicitly detailed.

The experiments were performed in constant assisting force mode. We have added this information to the text, the relevant section is copied here: “Upon adding 70-560 nM D-IX216 to a transcribing MtbRNAP (Figure 4a) under 18pN of assisting constant force, …”

9) P.6,L228-229 (and throughout the manuscript): The use of the word "region" does not seem appropriate. This really is a measure of polymerase "activity". There would seem to be no reason not to refer to these kinetic regions as fast, slow and super-slow (polymerase) activities.

We had a lot of trouble thinking about nomenclature for these. We like your suggestion and have changed them to ‘fast/slow/super-slow activity’, at least until we define the term ‘inhibited state’ later on.

10) P.6,L.250 and 252: Figure 4e should refer to the left and right panels, respectively, to focus on specific results.

We agree, this has been changed.

11) P.7,L.285-290: These sentences seem repetitive and are quite confusingly written. There must be a better way to express the differences in mathematical treatment of deep backtracking and shallow backtracking mechanics, and the relative fitting that describes the results.

We think the confusion results from including the final sentence, we think it is unnecessary and causes more confusion than clarity. The section now reads: “The recovery from the Stl-induced pauses before and after backtracking both exhibited single-exponential distributions (Figure 5c). A single-exponential distribution of backtrack durations is associated with *deep backtracking* and recovery via endonucleolysis, whereas a power law distribution (*t^-3/2^*) corresponds to diffusive recovery from more shallow backtracking (Figure 5d).”

12) P.8,L.307-308: These are not "seemingly conflicting" results. They are just "different results".

We have changed it to your suggestion; the text now reads “These differing results may reflect differences in the mechanism of Stl inhibition between EcoRNAP and MtbRNAP.”

13) P.8,L.331: Is this AAP's as a general class, or specifically D-IX216? I read it as the latter, but these continued confusing nomenclatures made simple reading of the manuscript quite difficult!

As noted before (comment 7), this is a typo. This should read D-IX216.

14) P.9,L.350: Two different concentrations of D-IX216 were used in the individual vs. combined inhibitor treatments with Stl. This difference makes these studies not exactly comparable and may change the results and interpretation. A repeat of these studies with comparable concentrations of drug is essential for this entire section and is a significant flaw in these studies.

We have now included data of D-IX216 at 560 nM alone to make the comparison with the combined case straightforward. The changes can be seen in Figure S5a, Figure 7, and in the passages that reference those figures.

15) P.9,L360: Rather than, "Therefore we propose", one might consider, "These studies appear consistent with the hypothesis …".

We agree, we have changed the text to the suggestion.

16) P.9,L389: "unbinding" should be replaced by "inhibitor dissociation".

We agree, it has been changed accordingly.

17) P.10,L.397-399: There must be a much simpler way of expressing the differences and relative value of single molecule studies vs. bulk studies? There is no question that the present studies identified new kinetic processes that would be impossible by bulk analysis. But the present sentence obscures this fact with repetitive words and pretentious language.

Actually, our intention here was not to compare bulk vs. single-molecule, as the referenced study is also a single-molecule force spectroscopy study. We at first thought it was possible that they did not document backtracking because they could not resolve it occurring (hence the reference to ‘spatiotemporal resolution’), but we realize it is also possible that the difference comes from the species of polymerase, as they studied EcoRNAP while we used MtbRNAP. The passage now reads: “Previous single-molecule studies have shown that Stl lengthens existing pauses in EcoRNAP, which is consistent with our observation that pause *c* is extended with Stl in MtbRNAP. However, in those studies, EcoRNAP was not found to backtrack after inhibition by Stl, which points to a difference in the inhibition mechanism of this drug with these two polymerases.”

18) P.10,L.412-415: It would seem to this referee that the final sentence could better reflect the capabilities of single molecule analysis for antibiotic mechanics in developing combined treatments for improved patient outcomes?

Yes, we have elaborated a bit and have expanded this final sentence to a small paragraph on the benefits of single-molecule approaches in drug discovery.

19) Figure 2d (right panel): Equations should be included in Supplementary Materials with terms for Boltzman relation defined (not presently included anywhere in the manuscript). This could include the actual calculations, which were not immediately obvious and required significant time to deconvolute.

We have added to Methods more description of the model, including an explanation of the form and the terms. We have added to the text a reference to this section when Figure 2d is cited. Also, the ‘simplifications’ (the other equations) were not defined and we believe actually to be more confusing than helpful, so we have removed them. It is now just one equation.

20) Figure 3b (right panel): Marking pause sites (a,b,c,d,his) would visually help readers.

Good idea, we have added those labels to the pauses.

21) Figure 3c (right panel): Marking pause site bands (a,b,c,d,his) for MtbRNAP would visually help readers. One assumes that the MtbRNAP pause site c is the major band and ultimately made up the MtbRNAP pause site array? Why was pause site b not used as an equivalent array? Seems a strong pause at least in early time points.

We thank the reader for this comment. Pause c is notably stronger than the other pauses, so we tried that first for creating the ruler and it worked so we did not test others. It is possible that pause b could have worked, too. The band of pause b indeed is initially as dark as pause c’s, which means they have similar pausing efficiencies (percentage of RNAPs that pause at that site), but the band lasts for much longer with pause c, meaning it has a longer pause duration (length of time the RNAP spends at this pause). We want the strongest pause (high efficiency and duration) for the ruler to make the detection of the ruler pause the easiest in the single-molecule assay.

We think maybe this could have come up since the band for pauses b and c for MtbRNAP might be ambiguous, we have now marked the relevant pause band on both sides of the gels to be more precise in showing which band corresponds to which pause. It should be more obvious that pause c is the strongest, and that the band lasts for much longer than e.g. pause b.

22) Figure 4a (left panel): What is the concentration of D-IX216?

It is 140nM, and it has now been noted in the figure.

23) Figure 4a (right panel): The pause dwell times for the fast kinetics prior to the super-slow activity seem different that the fast or slow activities?

In the single-molecule assay, activity from RNAP to RNAP can vary a bit, e.g. compare Figure 1a’s differing activities from the three different molecules. We did wonder if these speed and pausing variations in the region of fast activity before switches correlated to the activity in the slow or super-slow regions, but could not find any such correlation (see Figure 4—figure supplement 3c).

24) Figure 5e and 5g: The authors must use consistent nomenclature for pf, ef, and lp in these panels. Moreover, the meaning of AF (assisted force) and OP (opposing force) should be included in the legend.

We have changed the figures and text to refer to the states of the fast polymerase as pf/ef/lp. We have added the AF/OF definitions in the legend, as suggested.

25) Figure 6c: The illustration of polymerase kinetic activity and State 1,2,3 is different than Figure 4d. It would be useful if the authors stuck to a type of illustration to make comparisons of Figures easy for the reader.

We have formatted the two graphs to be the same, like Figure 4 (grouped by fast/slow/super-slow)

26) Suppl. Figure S2c: Use pf and ep, or some consistent nomenclature!

(Handled in comment 24)

27) Suppl. Figure S4c: Same as #26 – consistent nomenclature, please!

(Handled in comment 24)

Methods:1) The studies were not performed at physiological ionic strength (looks like ~70mN). One wonders if there might be significant kinetic differences.

The RNAP transcription experiments were performed in TB130 buffer which has 130 mM KCl and 10 mM MgCl_2_, which we believe to be in the physiological range. There are lower ionic strength buffers used, but those are for various preparatory steps (e.g. formation of the transcription complex is done in TB40: 40 mM KCl and 5 mM MgCl_2_).

Reviewer #2 (Recommendations for the authors):Referee report for "Pleomorphic effects of three small-molecule inhibitors on transcription elongation by *Mycobacterium tuberculosis* RNA polymerase" by Omar Herrera-Asmat et al.In this work, the authors use their well-established single-molecule optical tweezers assay to study the activity of the RNA polymerase from *Mycobacterium tuberculosis*. In particular, they investigate the changes in activity in the presence of three small-molecule inhibitors: N(α)-aroyl-N-aryl-phenylalaninamide (D-IX216), streptolydigin (Stl), and pseudouridimycin (PUM). They find a diverse range of effects on the polymerase, including transitions into a very slow activity state (induced by D-IX216 and PUM), induction of pausing and backtracking. Overall, the work provides interesting insights into the activity of an important pathogen with implications for the development of new drugs. This is a timely and relevant study that I found well-presented and interesting to read. Below, are some suggestions for clarifications and further improvements.- In several places they authors make statements like "there are no significant effects" (line 133) or "we observed an insignificant decrease" (line 307). It would be good to explicitly state what statistical test / method of analysis was used.

We thank the reviewer for this comment. We admit that we have been looser than we should have been with the term ‘significant’. For cases where we have not applied a significance test, we have changed language to not use that word. For example, in Line 133, we now say “no dramatic effects on k_pf_ or k_el_”. Due to the nature of the high number of transitions observed for each of these states, very slight differences can come up as statistically significant (narrow confidence intervals for fitting), our goal is to guide the reader towards the meaningful changes through the text.

- Related to the last point: In many of the figures there are bar charts essentially comparing various kinetic parameters. I believe it would be useful to -at least for key parameters- indicate which differences are statistically significant and which ones are not.

[We have addressed this point in the above response]

- The model used in Figure 2 to describe the force-dependent transcription velocity is not well explained. There is "a parameter A" (line 146), but then there is also lower case "a", which is never defined? Also, the authors give values for the stall force, which does not appear to be a parameter in the model, but do not seem to give values for Vmax, which is used to rescale the axes. I think the model should be better explained and all parameters defined. In addition, I am wondering whether it would be clearer and more informative to show velocity vs. force for the two polymerases, rather than to normalize both axis, which makes the plots look very similar.

We have changed how we depict the formula, and without the simplification (small a). We believe it is more understandable now, *v(f)=(1+A)/(1+Aexp(fδ/kT))*. We’ve updated the text and added more discussion in the Methods, too.

Regarding the second point (velocity vs. force in unnormalized units), this graph is generated by individually normalizing to each polymerase first, so it cannot be shown in unnormalized units. This matches what we and others have done previously in characterizing RNAP stall forces, see e.g. https://doi.org/10.1126/science.282.5390.902

- I believe it would be useful to (further) compare and contrast the present findings with previously published work on how different inhibitors affect RNA polymerases, see e.g. single-molecule work on SARS-CoV-2 (Seifert et al. eLife 2021; https://doi.org/10.7554/eLife.70968).

We thank the reviewer for this very good suggestion. We now have added a reference to that paper when discussing the effect of PUM, as the relevance to nucleotide analogs is important.